# Computational exploration of global venoms for antimicrobial discovery with Venomics artificial intelligence

Changge Guan [1,2,3,4,5], Marcelo D. T. Torres [1,2,3,4,5], Sufen Li[1,2,3,4] & Cesar de la Fuente-Nunez [1,2,3,4] ✉

The rise of antibiotic-resistant pathogens, particularly gram-negative bacteria, highlights the urgent need for novel therapeutics. Drug-resistant infections now contribute to approximately 5 million deaths annually, yet traditional antibiotic discovery has significantly stagnated. Venoms form an immense and largely untapped reservoir of bioactive molecules with antimicrobial potential. In this study, we mined global venomics datasets to identify new antimicrobial candidates. Using deep learning, we explored 16,123 venom proteins, generating 40,626,260 venom-encrypted peptides. From these, we identified 386 candidates that are structurally and functionally distinct from known antimicrobial peptides. They display high net charge and elevated hydrophobicity, characteristics conducive to bacterial-membrane disruption. Structural studies revealed that many of these peptides adopt flexible conformations that transition to α-helical conformations in membrane-mimicking environments, supporting their antimicrobial potential. Of the 58 peptides selected for experimental validation, 53 display potent antimicrobial activity. Mechanistic assays indicated that they primarily exert their effects through bacterial-membrane depolarization, mirroring AMP-like mechanisms. In a murine model of *Acinetobacter baumannii* infection, lead peptides significantly reduced bacterial burden without observable toxicity. Our findings demonstrate that venoms are a rich source of previously hidden antimicrobial scaffolds, and that integrating large-scale computational mining with experimental validation can accelerate the discovery of urgently needed antibiotics.

Drug-resistant infections account for approximately 5 million deaths annually worldwide[1], fueled by the rapid emergence of antibiotic-resistant pathogens. Among these, gram-negative bacteria, identified as priority pathogens by the World Health Organization (WHO), are particularly adept at developing resistance. Despite this growing threat, the development pipeline for novel antibiotics has stagnated over the past few decades due to high costs and lengthy timelines, emphasizing the urgent need for innovative therapeutic strategies[2,3].

One promising yet underexplored avenue lies in the vast molecular diversity of animal venoms, which have evolved over millions of

[1]Machine Biology Group, Departments of Psychiatry and Microbiology, Institute for Biomedical Informatics, Institute for Translational Medicine and Therapeutics, Perelman School of Medicine, University of Pennsylvania, Philadelphia, Pennsylvania, USA. [2]Departments of Bioengineering and Chemical and Biomolecular Engineering, School of Engineering and Applied Science, University of Pennsylvania, Philadelphia, Pennsylvania, USA. [3]Department of Chemistry, School of Arts and Sciences, University of Pennsylvania, Philadelphia, Pennsylvania, USA. [4]Penn Institute for Computational Science, University of Pennsylvania, Philadelphia, Pennsylvania, USA. [5]These authors contributed equally: Changge Guan, Marcelo D. T. Torres. ✉e-mail: cfuente@upenn.edu

years to interact with a wide range of biological targets[4–7]. These venoms are rich in bioactive peptides and proteins that exhibit diverse pharmacological effects, including antibacterial activity[8–12]. Venom-derived peptides offer several advantages over conventional small-molecule antibiotics. Unlike most traditional antibiotics, which target specific bacterial enzymes or biosynthetic pathways, many venom peptides act by disrupting bacterial membranes, a mechanism that bacteria struggle to evade through conventional resistance strategies[8–14]. In addition, venom peptides often exhibit broad-spectrum activity against both gram-positive and gram-negative bacteria, making them attractive candidates for combating multidrug-resistant pathogens. Venoms have already yielded therapeutic breakthroughs in other biomedical fields, with notable examples including ziconotide, marketed as Prialt®, an analgesic derived from cone snail venom used to treat chronic pain by selectively targeting voltage-gated calcium channels[15], and captopril, an antihypertensive agent originally sourced from snake venom[16]. However, the potential of venoms as a source of antimicrobial agents remains largely untapped, in part due to the challenges associated with systematically identifying bioactive peptides within complex venom proteomes.

Peptide-based antimicrobials represent an attractive alternative to both traditional antibiotics and protein-based therapeutics[17,18]. While small-molecule antibiotics are highly effective, they are often vulnerable to rapid resistance evolution, particularly when they act on single cellular targets. In contrast, proteins can achieve greater specificity but frequently suffer from issues related to stability, immunogenicity, and bioavailability[17]. Antimicrobial peptides, including those derived from venoms, offer a compelling middle ground by combining potent antimicrobial activity with structural flexibility and modifiability[19]. Their ability to be engineered for improved stability, selectivity, and pharmacokinetics makes them promising candidates for next-generation antimicrobial therapies.

Despite the clear potential of venom peptides[18,20,21], systematic discovery efforts have been limited by the sheer complexity of venom composition and the impracticality of high-throughput experimental screening. Recent advances in bioinformatics and machine learning have enabled the systematic mining of potential antimicrobial candidates from proteomes[22–29]. In this study, we applied APEX, a sequence-to-function deep learning-model[22,30], to systematically mine venom proteomes for antimicrobial candidates.

By leveraging neural network-based sequence encoding and activity prediction, APEX allows for in silico screening of thousands of venom-derived peptides, prioritizing candidates for experimental validation. This computational approach dramatically accelerates the identification of antimicrobial peptides, reducing reliance on resource-intensive biochemical assays.

Our study integrates computational discovery with experimental validation to uncover a set of venom-encrypted peptides (VEPs) with potent antimicrobial activity (Fig. 1a). We identified and synthesized 58 promising VEPs, testing their efficacy against multiple clinically relevant bacterial strains, including priority pathogens classified by the WHO. Beyond in vitro screening, we further demonstrate the translational potential of these peptides through preclinical validation in a murine model of *Acinetobacter baumannii* skin infection. Notably, our findings support the notion that venom-derived peptides not only retain their antimicrobial function when extracted from their parent toxins but also can serve as templates for future peptide-based therapeutics.

By combining machine learning, large-scale venom proteome analysis, and extensive experimental validation, this study establishes a framework for harnessing venom-derived peptides in antimicrobial drug discovery. Our findings highlight the rich and largely untapped potential of animal venoms in the fight against antibiotic resistance and lay the groundwork for future research into computationally-guided peptide discovery.

## Results

### Mining venoms for antimicrobials

We sourced venom proteins from four databases: ConoServer (focusing on conopeptides, Supplementary Fig. 1)[31], ArachnoServer (spider proteins, Supplementary Fig. 2)[32], ISOB (indigenous snake proteins, Supplementary Fig. 3)[33], and VenomZone (covering six taxa: snakes, scorpions, spiders, cone snails, sea anemones, and insects, Supplementary Fig. 4)[34]. The VenomZone dataset, curated from UniProtKB, was represented in our study by UniProt. Altogether, we compiled 16,123 venom proteins, which were computationally truncated (Supplementary Fig. 5) and processed to generate 40,626,260 VEPs.

To analyze differences across the databases, we performed a species overlap analysis (Fig. 1b). UniProt contained the largest number of unique species (699), reflecting its extensive coverage. Conoserver and Arachnoserver encompassed smaller unique subsets (16 and 12, respectively), while ISOB contained no unique species. These results highlight the complementary nature of these databases, emphasizing the value of integrating multiple sources to achieve comprehensive venom protein diversity.

Using APEX, a deep learning model, we predicted bacterial strain-specific MIC values for each peptide and used the median MIC as a measure of antimicrobial activity. We identified 7,379 VEPs with a median MIC $\leq 32\ \mu\mathrm{mol\ L}^{-1}$ (Supplementary Data 1). Further filtering criteria (see Methods: Venom-encrypted peptide selection) based on sequence similarity to known antimicrobial peptides (AMPs) yielded 386 candidates with low similarity to existing AMPs (Supplementary Table 1 and Supplementary Data 2).

To visualize sequence diversity, we compared the 386 VEPs with 19,762 known AMPs from the DBAASP database. Pairwise alignment (see Methods: Peptide sequence similarity) and uniform manifold approximation and projection (UMAP) revealed that most known AMPs clustered densely, reflecting a high sequence similarity matrix (Fig. 1c).

Most known AMPs formed a dense cluster, indicating high sequence similarity, with a minority scattered outside this cluster, representing more diverse sequences. VEPs derived from ConoServer and ArachnoServer tended to cluster closer to known AMPs, reflecting relatively higher sequence similarity. In contrast, UniProt-derived VEPs mapped farther from the AMP cluster, partially overlapping with scattered AMPs and occupying previously unexplored regions of sequence space. ISOB-derived VEPs were the most distant from known AMPs, forming isolated clusters that represent a promising source of AMP sequences (Fig. 1c).

To determine whether VEPs with low sequence similarity to known AMPs share key physicochemical characteristics, we analyzed their distribution in physicochemical feature space (Supplementary Fig. 6). While known AMPs from DBAASP clustered centrally, UniProt-derived VEPs formed three distinct clusters, Arachnoserver-derived VEPs formed two clusters, and ISOB and Conoserver each formed one cluster. UniProt cluster overlapped with ConoServer, while the ISOB-derived cluster remained entirely isolated. UniProt- and Arachnoserver-derived clusters that did not overlap with known AMPs represent unexplored regions of sequence space (Fig. 1c).

These findings suggest that our approach identifies both AMP-like peptides that differ in sequence while sharing similar physicochemical properties and entirely different AMP families that deviate in both sequence and characteristics.

### Composition and physicochemical features

A comparison of amino acid composition between VEPs and DBAASP AMPs revealed distinct profiles (Fig. 1d and Supplementary Fig. 7). VEPs had lower cysteine, aspartic acid, histidine, and isoleucine, while showing higher phenylalanine, lysine, and arginine content. ISOB-derived VEPs were particularly enriched in phenylalanine, whereas Conoserver-derived VEPs displayed pronounced arginine content.

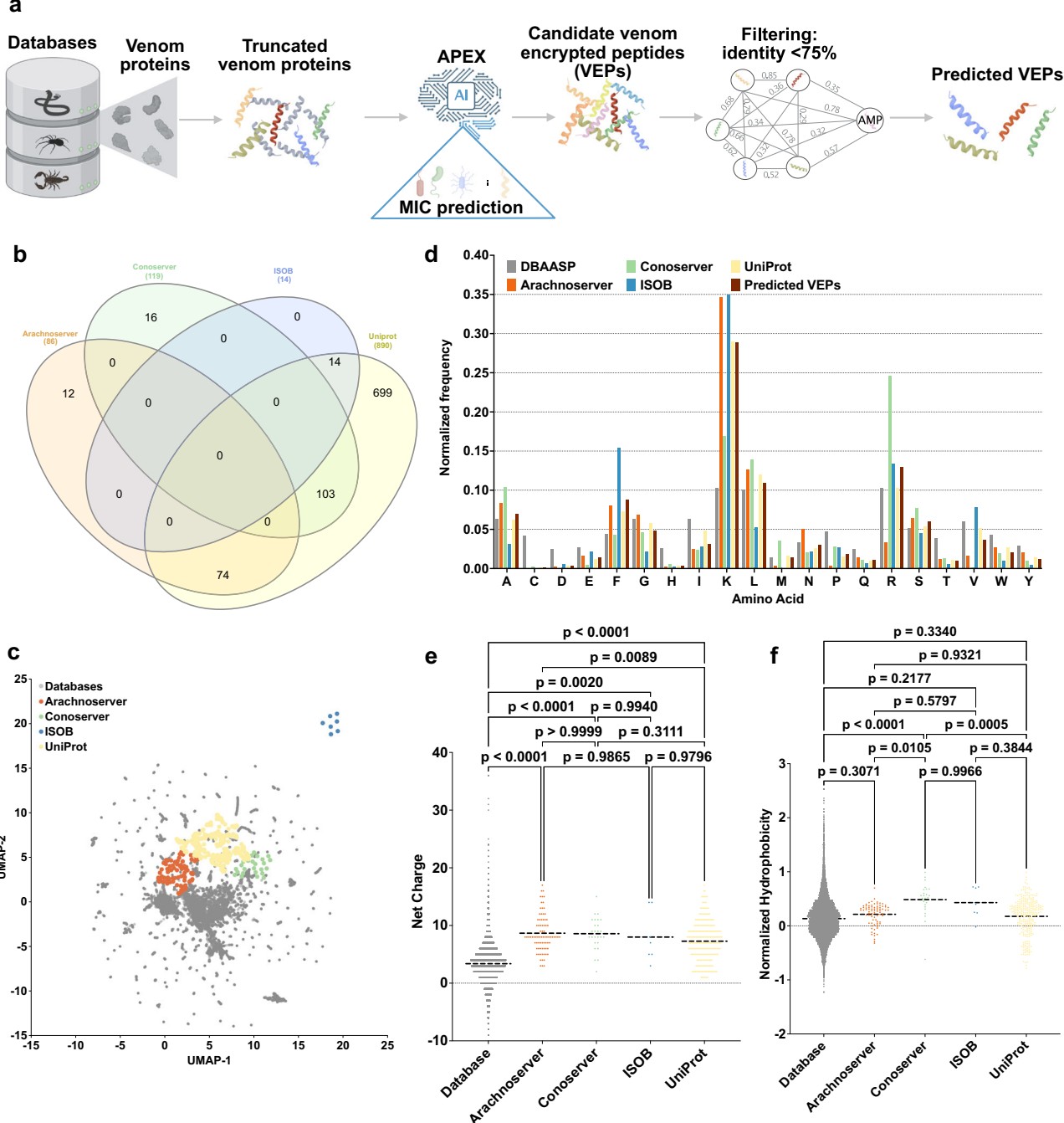

**Fig. 1 | Exploration of global venoms for antimicrobial discovery. a** Mining framework for AMPs. Our framework employs a three-stage approach to identify AMP candidates from venom proteins. Initially, a peptide library is generated using a sliding window method, extracting peptides ranging from 8 to 50 amino acid residues in length. Subsequently, minimum inhibitory concentration (MIC) values of peptides against bacterial strains were predicted by APEX. Finally, candidate VEPs are selected based on sequence similarity, yielding a set of unique and potent molecules. **b** A Venn diagram illustrating species overlap among the four databases used as a source of venom proteins. Species names extracted from these databases were analyzed to identify diversity. **c** Physicochemical feature space exploration. The graph illustrates a bidimensional sequence space visualization of peptide sequences found in DBAASP and antimicrobial venom-derived EPs (VEPs) discovered by APEX in venom proteins from multiple source organisms. The physicochemical features were calculated for peptide sequences, which was made up of the feature vector for representing peptides. Each row in the matrix represents a feature representation of a peptide based on its amino acid composition. Uniform manifold approximation and projection (UMAP) was applied to reduce the feature representation to two dimensions for visualization. **d** Comparison of amino acid frequency in VEPs with known antimicrobial peptides (AMPs) from database. Distribution of two physicochemical properties for peptides with predicted antimicrobial activity, compared with AMPs from databases: **e** net charge and **f** normalized hydrophobicity. Net charge influences the initial electrostatic interactions between the peptide and negatively charged bacterial membranes, while hydrophobicity affects interactions with lipids in the membrane bilayers. Statistical significance in (**e**, **f**) was determined using two-tailed *t*-tests followed by the Mann–Whitney test; *p* values are shown in the graph. The solid line inside each box represents the mean value for each group. Panel (**a**) created in BioRender. De la Fuente-Nunez, C. (2025) https://BioRender.com/es1g25g.

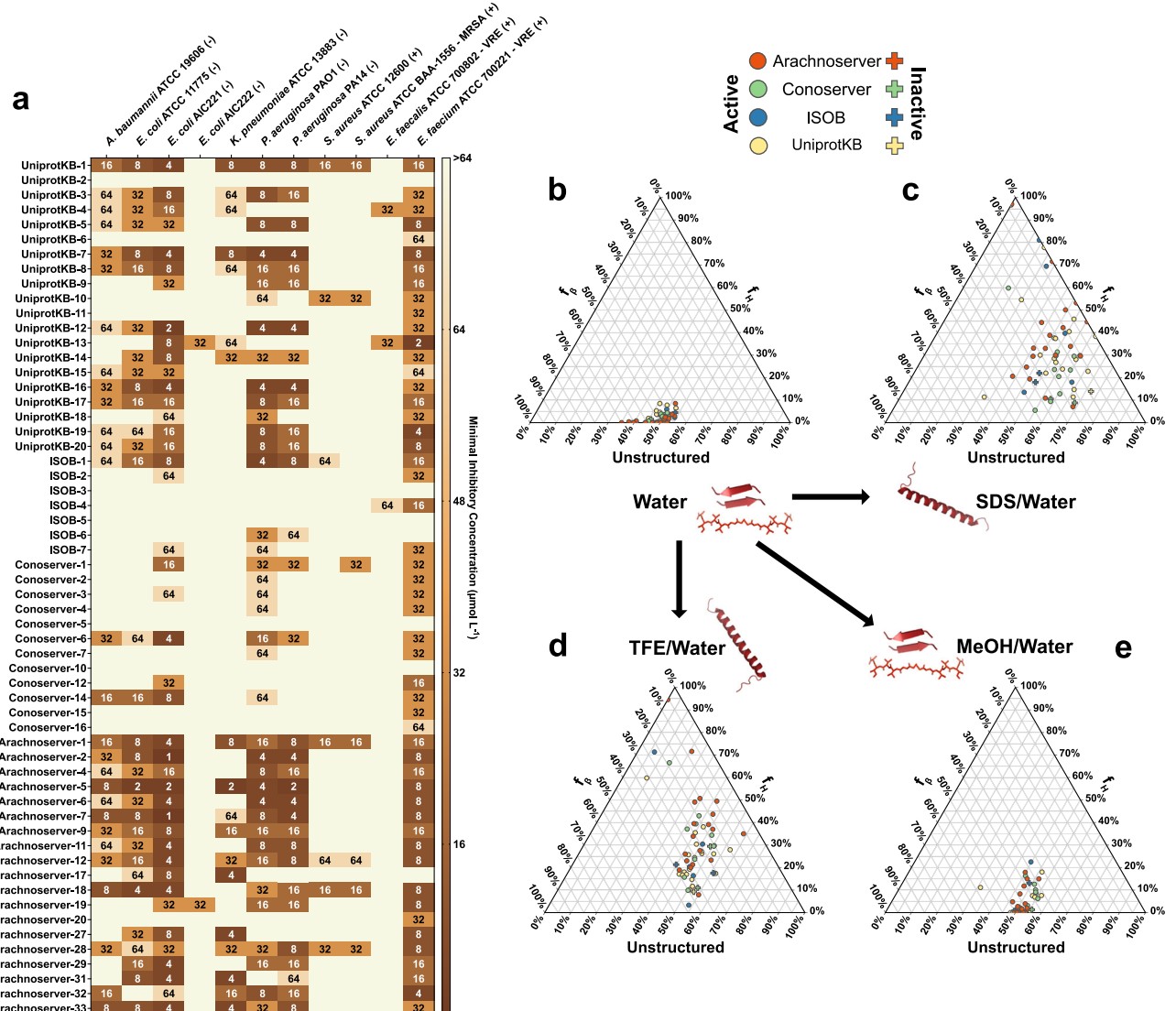

**Fig. 2 | Antimicrobial activity and secondary structure profiles of antimicrobials from venoms. a** Heatmap displaying the antimicrobial activities (µmol L⁻¹) of active antimicrobial agents from venoms against 11 clinically relevant pathogens, including antibiotic-resistant strains (gram-negative and gram-positive bacteria as indicated as such by – and + succeeding their names). Briefly, 10⁵ bacterial cells were incubated with serially diluted VEPs (1–64 µmol L⁻¹) at 37 °C. Bacterial growth was assessed by measuring the optical density at 600 nm in a microplate reader one day post-treatment. The MIC values presented in the heatmap represent the mode of the replicates for each condition. **b**–**e** Ternary plots showing the percentage of secondary structure for each peptide (at 50 µmol L⁻¹) in four different solvents: **b** water, **c** Sodium dodecyl sulfate (SDS, 10 mmol L⁻¹) in water, **d** 60% trifluoroethanol (TFE) in water, and **e** 50% methanol (MeOH) in water. Secondary structure fractions were calculated using the BeStSel server[39]. Circles indicate active VEPs, while crosses represent inactive peptides.

Notably, Arachnoserver- and ISOB-derived VEPs were enriched in lysine.

To further understand the physicochemical properties contributing to antimicrobial activity, we benchmarked VEPs against known AMPs (Fig. 1e, f and Supplementary Fig. 8). VEPs were generally more positively charged, facilitating electrostatic interactions with the negatively charged bacterial membranes[22]. They alsbicity, likely driven by their increased phenylalanine and arginine content. In ISOB- and Conoserver-derived VEPs, these features enhanced amphiphilicity (Supplementary Fig. 8a), promoting secondary structure formation and membrane-associated activity.

Additionally, VEPs displayed higher isoelectric points than known AMPs (Supplementary Fig. 8b), consistent with their elevated cationic residue content. By design, the APEX model excluded peptides with high cysteine content, thereby avoiding many Conoserver-derived peptides rich in disulfide bridges. Despite their elevated phenylalanine levels, VEPs maintained comparable normalized hydrophobic moments (Supplementary Fig. 8c) and aggregation propensities (Supplementary Fig. 8d) to conventional AMPs, with amphiphilic distribution likely mitigating hydrophobic clustering.

Collectively, these results delineate the unique composition and physicochemical properties of VEPs, highlighting their potential as promising antimicrobial candidates.

## Antimicrobial activity assays

Among the 58 VEPs tested, 53 (91.4%) exhibited activity against at least one pathogenic strain. Notably, all Arachnoserver-derived peptides were active, emphasizing their strong antimicrobial potential (Fig. 2a). In contrast, some UniProt-derived VEPs (from VenomZone) demonstrated limited potency: UniprotKB-2 showed no activity, while UniprotKB-6 and UniprotKB-11 were active only against *Enterococcus faecium*.

The inactive or minimal activity of UniProtKB-2, -6, and -11 was associated with lower hydrophobicity and net charge, underscoring

the important role of these parameters in facilitating membrane interactions. Conversely, ISOB-derived VEPs with enhanced normalized hydrophobicity exhibited improved antimicrobial performance. Among Conoserver-derived VEPs, an intermediate balance of hydrophobicity and net charge appeared to be optimal for activity. In Arachnoserver-derived VEPs, where all candidates were active, efficacy seemed to be driven by sequence-specific features rather than general physicochemical properties.

These findings underscore the importance of physicochemical characteristics, such as charge and hydrophobicity, in effective bacterial membrane disruption while also highlighting the significant role of sequence-specific factors in determining antimicrobial efficacy.

### Secondary structure studies

The secondary structure of short peptides is inherently dynamic, often transitioning between disordered and ordered conformations depending on the surrounding environment, particularly at hydrophobic/hydrophilic interfaces. These structural transitions are critical for defining the biological functions of peptides, including their antimicrobial activity.

To investigate the structural behavior of the synthesized VEPs, we performed circular dichroism (CD) spectroscopy in diverse environments: water, sodium dodecyl sulfate (SDS)/water (10 mmol L$^{-1}$), methanol (MeOH)/water (1:1, v:v), and trifluoroethanol (TFE)/water (3:2, v:v). Each medium was chosen to simulate specific physicochemical conditions relevant to peptide behavior. SDS micelles mimic biological lipid bilayers, offering a membrane-like environment conducive to evaluating interactions with bacterial membranes[35]. The TFE/water mixture is a known helical-inducer that promotes intramolecular hydrogen bonding by dehydrating peptide backbone amide groups, thereby favoring α-helical conformations[36,37]. Conversely, the MeOH/water mixture promotes interchain hydrogen bonding, stabilizing β-like structures, while hydrophobic side chains cluster to minimize contact with water, enhancing β-like conformations[38].

CD spectra were recorded for all VEPs at 50 µmol L$^{-1}$ over a wavelength range of 260 to 190 nm (Supplementary Fig. 9a–d). The beta structure selection (BeStSel) algorithm was employed to deconvolute the spectra and quantify the secondary structure content[39] (Fig. 2b–e). As expected for short peptides (<50 amino acid residues), VEPs were predominantly unstructured in water (Fig. 2b and Supplementary Fig. 9a, e), though with a slight propensity toward β-like conformations ($f_\beta < 45\%$; Supplementary Fig. 9e). A similar trend was observed in the β-inducing medium (MeOH/water), where the β-content modestly increased (Fig. 2e and Supplementary Fig. 9d, e).

In contrast, VEPs exhibited a pronounced structural transition in SDS micelles (Fig. 2c and Supplementary Fig. 9c, e) and TFE/water mixture (3:2, v:v; Fig. 2d and Supplementary Fig. 9b, e), adopting α-helical conformations. This shift from disordered to α-helical structures highlights their responsiveness to membrane-mimicking environments and helical-inducing media, consistent with typical behavior observed for antimicrobial peptides[6,40,41].

Interestingly, this behavior distinguishes VEPs from other classes of encrypted peptides, including those predicted by earlier proteome mining using APEX[22], which predominantly adopted unstructured or β-like conformations, even in membrane-like or helical-inducing environments. Similarly, small open reading frame-encoded peptides (SEPs) and bacterial proteome-derived encrypted peptides[42,43] showed limited helical propensity under comparable conditions. Instead, VEPs exhibited a structural response more akin to archaeasins, which also demonstrated a clear transition to α-helical conformations in helical-inducing media and upon interacting with lipid bilayers[30]. These findings suggest that VEPs may be uniquely suited for membrane-associated functions, likely contributing to their observed antimicrobial efficacy.

### Mechanism of action studies

To investigate whether VEPs exert their activity via membrane-related mechanisms, we evaluated their effects on bacterial outer and cytoplasmic membranes using fluorescence assays. We used 1-(N-phenylamino)-naphthalene (NPN) assays to assess bacterial outer membrane permeabilization (Fig. 3a). Among the peptides tested, 23 VEPs effectively permeabilized the outer membrane. Notably, Arachnoserver-18, derived from the protein M-oxotoxin-Ot2d of the spider *Oxyopes takobius*; ConoServer-6, derived from the protein Bt211 precursor, a widely studied conotoxin from the betuline cone (*Conus betulinus*); and ConoServer-7, derived from the protein Con-ins G1b precursor of *Conus geographus*, a cone snail known for having the most potent venom among the *Conus* genus[44], showed superior permeabilization activity. Polymyxin B and levofloxacin were used as controls in these experiments[24]. Overall, VEPs demonstrated permeabilization comparable to or better than other AMPs[7,45,46] or other human- or animal-derived EPs[22,24].

We next evaluated cytoplasmic membrane depolarization using 3,3′-dipropylthiadicarbocyanine iodide (DiSC$_3$-5), a fluorophore that detects membrane potential changes. Among the 28 peptides tested against *P. aeruginosa* PAO1, 26 VEPs depolarized the cytoplasmic membrane more effectively than the control groups treated with polymyxin B and levofloxacin (Fig. 3b)[24]. However, the depolarization efficacy of VEPs was less pronounced compared to other peptide families[42], such as those derived from archaeal proteomes (archaeasins)[30] and SEPs[42]. Against the gram-positive bacterium *S. aureus*, VEPs exhibited slightly better depolarization activity than *P. aeruginosa* (Fig. 3c), though their performance remained below that of other reported peptide depolarizers[23,43].

These findings suggest that VEPs primarily exert their antimicrobial effects through cytoplasmic membrane depolarization rather than outer membrane permeabilization. This mode of action aligns with that of some AMPs[45,46] and EPs[24] but differs from certain computationally predicted peptides[42].

### In vitro cytotoxicity of VEPs

Cytotoxicity was assessed using human embryonic kidney (HEK293T) cells. Some VEPs, especially from the UniprotKB and Arachnoserver datasets, were cytotoxic at CC$_{50} \leq 64$ µmol L$^{-1}$ (Fig. 4a), mirroring their potent antimicrobial activity. To further test the toxicity of these molecules, we performed hemolysis assays by exposing the peptides to human red blood cells (Supplementary Fig. 11). In general, the peptides were not as active as against HEK293T cells, and only the most active ones presented some degree of hemolytic activity. VEPs from UniprotKB, ISOB, and Conoserver datasets were not toxic. A few Arachnoserver VEPs showed hemolytic activity and were not considered for further in vivo experimental validation. These findings underscore the importance of fine-tuning VEP properties to balance antimicrobial efficacy with reduced cytotoxicity, guiding further peptide optimization.

### Anti-infective activity in preclinical animal models

To determine the in vivo efficacy of lead VEPs, we used a skin abscess mouse model infected with *A. baumannii*, a clinically significant pathogen (Fig. 4b). Based on their selectivity index (Supplementary Table 3), we selected the most active VEPs with low toxicity. Three VEPs demonstrated promising activity: UniProtKB-7, derived from the Im-1 toxin of the scorpion *Isometrus maculatus*; ConoServer-14, derived from the Elevenin-Vc1 protein of the cone snail *Conus quercinus*; and Arachnoserver-5, derived from the M-lycotoxin-Gri2c protein of the wolf spider *Geolycosa riograndae*.

A single topical dose of each VEP at its MIC significantly reduced bacterial counts 2 days post-infection. Arachnoserver-5 achieved a two-log reduction in bacterial load, comparable to the activity of polymyxin B and levofloxacin controls. Four days post-infection, all

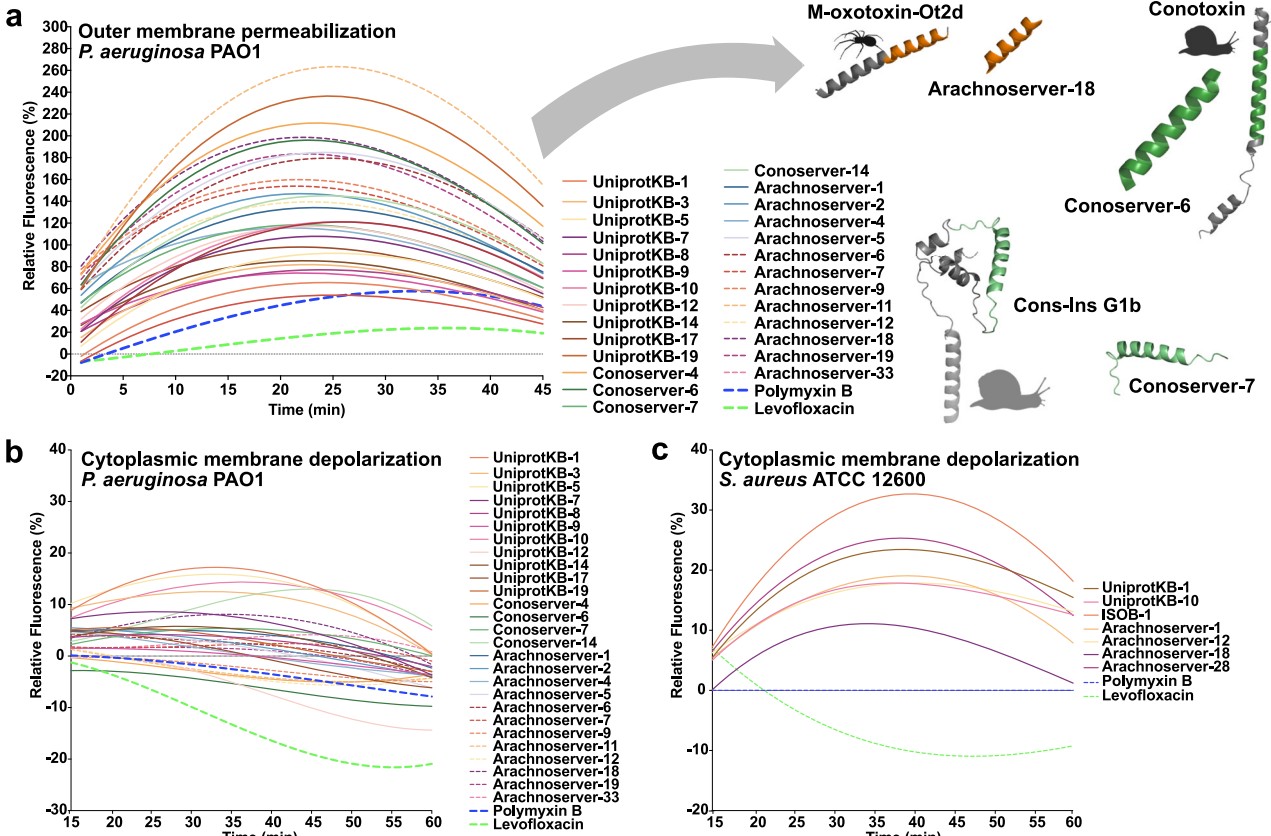

**Fig. 3 | Mechanism of action of antimicrobials from venoms.** To assess whether VEPs act on bacterial membranes, all active peptides against *P. aeruginosa* PAO1 were subjected to outer membrane permeabilization, and peptides active against *P. aeruginosa* PAO1 and *S. aureus* ATCC 12600 were tested in cytoplasmic membrane depolarization assays. The fluorescent probe 1-(N-phenylamino)naphthalene (NPN) was used to assess membrane permeabilization (**a**) induced by the tested VEPs in *P. aeruginosa* PAO1 cells. The fluorescent probe 3,3′-dipropylthiadicarbocyanine iodide (DiSC$_3$-5) was used to evaluate membrane depolarization of **b** *P. aeruginosa* PAO1 or **c** *S. aureus* ATCC 12600 caused by VEPs. The values displayed represent the relative fluorescence of both probes, with nonlinear fitting compared to the

baseline of the untreated control (buffer + bacteria + fluorescent dye) and benchmarked against the antibiotics polymyxin B and levofloxacin. All experiments were performed in three independent replicates. The relative fluorescence values in (**a**–**c**) were calculated as the percentage difference between the sample and the untreated control using Eq. 3 (Methods). The untreated control (buffer + bacteria + fluorescent dye) served as the baseline, and polymyxin B was used as a positive control for benchmarking. The protein and peptide structures depicted in the figure were created with PyMOL Molecular Graphics System, version 3.1, Schrödinger, LLC. Panel (**a**) created in BioRender. De La Fuente-Nunez, C. (2025) https://BioRender.com/1bfl5da.

---

three VEPs continued to suppress bacterial growth, with Arachnoserver-5 producing a three-log reduction relative to untreated controls (Fig. 4c). Importantly, no significant changes in body weight were observed in treated animals, indicating minimal toxicity under these conditions (Supplementary Fig. 12).

## Discussion

This work demonstrates how large-scale, machine-learning exploration of venom proteomes—coupled with focused experimental validation—can unlock an untapped source of antibiotics. The VEPs identified in this work exhibit distinct sequences and physicochemical properties, retain membrane-active mechanisms characteristic of known antimicrobial peptides, and demonstrate promising antimicrobial activity in both in vitro assays and preclinical animal models.

Our findings highlight the power of combining digital data with machine learning to accelerate antibiotic discovery, building on advances in this emerging field[22–24,29,47].

Future work will focus on performing targeted chemical modifications of these venom-derived peptides to further enhance their stability, bioavailability, and selectivity. These optimization efforts aim to maximize the therapeutic potential of VEPs as next-generation antibiotics.

While APEX has proven effective in accelerating the discovery of antimicrobials, several limitations remain. One constraint is its reliance

on discrete MIC values, which are recorded in multiples of 2, and the exclusive use of AAindex features, potentially limiting prediction accuracy and generalizability. Another limitation is the lack of interpretability in APEX's predictions, as it does not identify specific sequence features responsible for antimicrobial activity. While including all 34 strains could have provided additional computational insights, experimental validation of a larger number of strains will be pursued in future studies.

We acknowledge that the VEPs identified present high net positive charge and hydrophobicity, which can potentially be associated with toxicity. Nevertheless, here we show that several of the lead candidates were not toxic against human red blood cells and human cell lines. In addition, those features also play crucial roles in antimicrobial activity and are thus needed to yield active compounds.

Moreover, venom-derived compounds may possess toxicity via modulation of ion channels. Our predictions indicate that nearly 40% of the identified VEPs were not expected to affect potassium channels (Supplementary Table 4). However, ~60% of the peptides, including those tested in vivo, were predicted to modulate ion channels, highlighting the need for further experimental validation. Future studies should incorporate electrophysiological assays to directly evaluate the effects of VEPs on specific ion channels, particularly since systemic administration or prolonged topical use may cause unintended side effects. Additional optimization may also be required to reduce

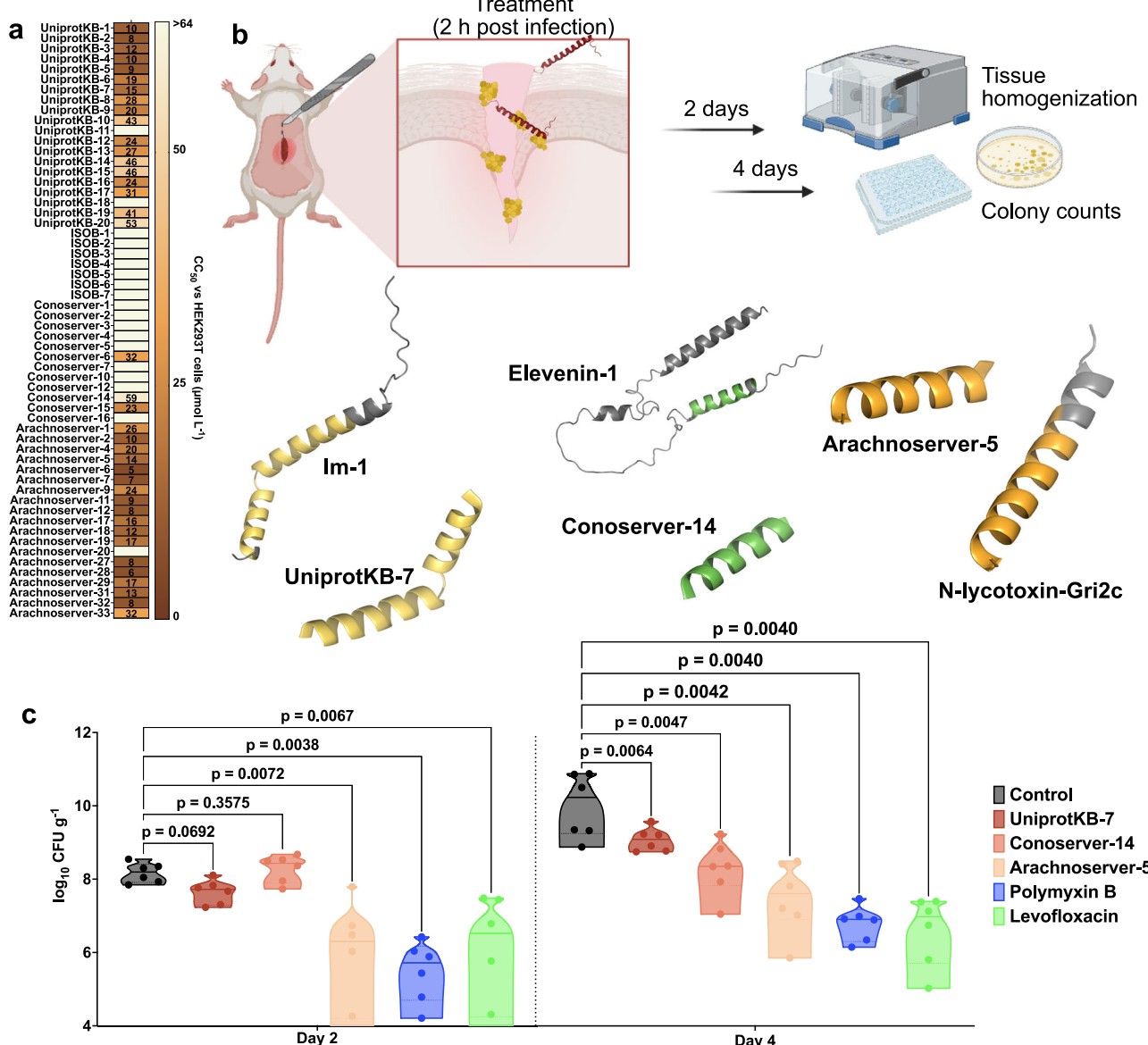

**Fig. 4 | Cytotoxic and anti-infective activity of antimicrobials from venoms.**
**a** Heatmap exhibiting the cytotoxic concentrations leading to 50% cell lysis ($CC_{50}$) in human embryonic kidney (HEK293T) cells, determined by interpolating dose-response data using a nonlinear regression curve. All experiments were performed in three independent replicates. **b** Schematic representation of the skin abscess mouse model used to assess the anti-infective activity of VEPs ($n = 6$) against *A. baumannii* ATCC 19606. **c** UniprotKB-7, conoserver-14, and arachnoserver-5 were administered at their MIC in a single dose 2 h post-infection. Arachnoserver-5

inhibited the proliferation of the infection for up to 4 days after treatment compared to the untreated control group, at levels comparable to the control antibiotics, polymyxin B and levofloxacin. To determine statistical significance in (**c**), one-way ANOVA followed by Dunnett's test was employed, and the respective *p* values are presented for each group. All groups were compared with the untreated control, and the violin plots display the median and upper and lower quartiles. Panel (**b**) created in BioRender. De la Fuente-Nunez, C. (2025) https://BioRender.com/ybengo5.

potential toxicity. Expanding the panel of bacterial strains tested would further enhance our computational-experimental framework by enabling feedback-driven refinement of the predictive model.

To address these limitations and enhance APEX's utility, several strategies can be implemented. First, incorporating self-attention layers or related explainability tools could identify sequence motifs that drive activity, enhancing interpretability and guiding rational design. Second, data-augmentation strategies and larger, more diverse training sets, including negative examples, should broaden model scope. Employing data-augmentation techniques could enhance generalizability across diverse peptide datasets. Third, integrating large language models could capture complex sequence relationships, further improving prediction accuracy and broadening APEX's applicability.

In addition to these computational refinements, several experimental considerations will guide future work. Although LB medium was used in this study for initial peptide screening, future studies will incorporate standardized media such as Mueller-Hinton broth, in alignment with international guidelines for antimicrobial susceptibility testing. The addition of compounds like Tween-80 to minimize peptide adhesion to plastic surfaces will also be considered. Furthermore, while topical application limits systemic exposure, the potential for ion channel modulation, particularly of potassium channels, underscores the need for electrophysiological assays to assess toxicity, especially if systemic administration is envisioned. These steps will further strengthen the development and evaluation of venom-derived peptides as therapeutic candidates.

## Methods

### Ethics statement

All animal experiments were conducted in accordance with the guidelines established by the Institutional Animal Care and Use Committee (IACUC) at the University of Pennsylvania. All procedures were reviewed and approved by University Laboratory Animal Resources (ULAR) under protocol number 806763.

### Encrypted peptides in venom proteomes

The venom protein sequences were collected from https://www.snakebd.com/ (Snakes), https://arachnoserver.qfab.org/mainMenu.html (Spider), https://www.conoserver.org/ (Carnivorous marine cone snails), and https://venomzone.expasy.org/ (Venom Zone) (accessed data: August 30th, 2023). About 654, 2206, 5494, and 7769 proteins were obtained from the above four databases, respectively. Venom protein substrings ranging from 8 to 50 amino acids in the sequences, comprising only canonical amino acids, were considered as the venom-encrypted peptides (VEPs). The venom proteins were pre-processed in three ways based on length: (1) no truncation for lengths ≤8; (2) truncation using a sliding window (range from 8 to maximum sequence length) for lengths between 8 and 50; (3) truncation using a sliding window (range from 8 to 50) for lengths >50. In total, 40,626,260 VEPs were obtained from venom protein sequences, which were for further study.

### APEX

APEX is a bacterial strain-specific antimicrobial activity predictor[22], and was trained on an in-house peptide dataset and publicly available antimicrobial peptides (AMPs) from DBAASP[48]. Specifically, APEX is a multiple-target tasks model that can predict minimum inhibitory concentrations (MICs) values of peptides against 34 bacterial strains[22]. Python 3.9 is required to set up APEX's environment (https://gitlab.com/machine-biology-group-public/apex/-/blob/main/README.md). Next, install PyTorch version 1.11.0 with CUDA support by running the following command:

pip install torch==1.11.0+cu113 torchvision==0.12.0+cu113 torchaudio==0.11.0 --extra-index-url https://download.pytorch.org/whl/cu113

After that, install the required dependency libraries using this command:

pip install numpy==1.23 scipy==1.10 matplotlib==3.9.4 pandas==2.2.3 scikit-learn==1.6.1 rdkit==2024.3.2

This setup will ensure that all necessary packages are properly installed for running APEX. To predict the sequence, prepare a text file (for example: test_seqs.txt) in which each line contains one candidate sequence. Then run the command:

python predict.py test_seqs.txt

The output will be a.csv file with MIC predictions organized by sequence and strain. This file can be used to analyze the peptides according to any requirements.

### Venom-encrypted peptide selection

APEX was used to predict the antimicrobial activity for the 40,626,260 encrypted peptides derived from the venom proteome. Peptides with less than eight amino acid residues often lack the necessary amphipathicity and charge balance required for membrane interaction, thus those were excluded from the selection process[49,50]. We used the mean MIC value against the eleven pathogen strains to rank and select the encrypted peptides for chemical synthesis and experimental validation. When selecting the peptides, we also make sure they met the following criteria:

1. The selected peptide should have ≤32 µmol L$^{-1}$ median MIC by prediction.
2. The selected peptide should have <75% sequence similarity to all in-house peptides and publicly available AMPs.

3. The selected peptides themselves should have <75% sequence similarity.

After all filters, we performed an extra selection step where synthesis feasibility and aggregation propensity were taken into account.

### Physicochemical property analysis

The twelve physicochemical properties of peptides, including normalized hydrophobic moment, normalized hydrophobicity, net charge, isoelectric point, penetration depth, tilt angle, disordered conformation propensity, linear moment, propensity to aggregation in vitro, angle subtended by the hydrophobic residues, amphiphilicity index, and propensity to PPII coil, were obtained from the DBAASP server[48]. Note that the Eisenberg and Weiss scale[51] was chosen as the hydrophobicity scale.

### Phylogenetic tree visualization

To obtain the phylogenetic tree, the taxon IDs of organisms obtained from four databases were uploaded to the NCBI Taxonomy Common Tree (https://www.ncbi.nlm.nih.gov/Taxonomy/CommonTree/wwwcmt.cgi). The resulting tree file from NCBI was then visualized via iTOL (https://itol.embl.de/).

### Peptide sequence similarity

We applied the Needleman–Wunsch algorithm in the function "needleall" from the EMBOSS software package (version 6.6.0.0)[52] to estimate the similarity between our VEP with median MIC ≤32 µmol L$^{-1}$ and AMPs in the DBAASP dataset. The parameters used are all default, and the parameter 'identity' was sifted out for the graph.

### AA frequencies calculation

The function "ProtParam.ProteinAnalysis" was imported from the Biopython module "Bio.SeqUtils.ProtParam" (version 1.75)[53], which was used to count the total number of amino acids in a protein sequence and calculate the percentage composition of each amino acid in a protein sequence for two-level analysis, including amino acid level and sequence level.

Amino acid level (Eq. 1):

$$AA_i = \frac{\sum_{j=1}^{n} aa_{ij}}{\sum_{i}^{20} \sum_{j=1}^{n} aa_{ij}} \tag{1}$$

Where $aa_{ij}$ is the number of amino acids i in sequence j and $AA_i$ is the frequency of amino acid I. $n$ is the total number of sequences and 20 is the total number of amino acids.

Sequence level (Eq. 2):

$$AA_i = \frac{\sum_{j=1}^{n} aa_{ij}}{n} \tag{2}$$

Where $aa_{ij}$ is the frequency of amino acid i in sequence j and $AA_i$ is the frequency of amino acid i. $n$ is the total number of sequences.

### Peptide sequence space visualization

Given a peptide dataset, a similarity matrix containing the pairwise peptide sequence similarity could be calculated by previous method (peptide sequence similarity). Uniform manifold approximation and projection (UMAP) was then used to transform the similarity matrix into a two-dimensional space. We used this space as a proxy for the peptide sequence space and visualized the peptides' distribution/spread/location in it.

## Ion channel modulation predictions

The ability of peptides modulating four ion channels, including (sodium ion channel, potassium ion channel, calcium ion channel, and nAChRs) were predicted using the deep learning model PrIMP with multi-task learning by using the default parameter [14].

## Peptide synthesis

All peptides used in the experiments were purchased from AAPPTec and synthesized by solid-phase peptide synthesis using the Fmoc strategy.

## Bacterial strains and growth conditions

In this study, we used the following pathogenic bacterial strains: *Acinetobacter baumannii* ATCC 19606, *Escherichia coli* AIC221 [*Escherichia coli* MG1655 phnE_2::FRT (control strain for AIC222)] and *Escherichia coli* AIC222 [*Escherichia coli* MG1655 pmrA53 phnE_2::FRT (polymyxin resistant; colistin-resistant strain)], *Klebsiella pneumoniae* ATCC 13883, *Pseudomonas aeruginosa* PAO1, *Pseudomonas aeruginosa* PA14, *Staphylococcus aureus* ATCC 12600, methicillin-resistant *Staphylococcus aureus* ATCC BAA-1556, vancomycin-resistant *Enterococcus faecalis* ATCC 700802, and vancomycin-resistant *Enterococcus faecium* ATCC 700221. Pseudomonas Isolation (*Pseudomonas aeruginosa* strains) agar plates were exclusively used in the case of *Pseudomonas* species. All the other pathogens were grown in Luria-Bertani (LB) broth and on LB agar. In all the experiments, bacteria were inoculated from one isolated colony and grown overnight (16 h) in liquid medium at 37 °C. In the following day, inoculums were diluted 1:100 in fresh media and incubated at 37 °C to mid-logarithmic phase.

## Minimal inhibitory concentration assays

Broth microdilution assays were performed to determine the minimum inhibitory concentration (MIC) values of each peptide. Peptides were added to nontreated polystyrene microtiter 96-well plates and twofold serially diluted in sterile water from 1 to 64 μmol L$^{-1}$. Bacterial inoculum at $2 \times 10^6$ CFU mL$^{-1}$ in LB medium was mixed 1:1 with the peptide. The MIC was defined as the lowest concentration of peptide able to completely inhibit bacterial growth after 24 h of incubation at 37 °C. All assays were done in three independent replicates.

## Circular dichroism experiments

The circular dichroism experiments were conducted using a J1500 circular dichroism spectropolarimeter (Jasco) in the Biological Chemistry Resource Center (BCRC) at the University of Pennsylvania. Experiments were performed at 25 °C, the spectra graphed are an average of three accumulations obtained with a quartz cuvette with an optical path length of 1.0 mm, ranging from 260 to 190 nm at a rate of 50 nm min$^{-1}$ and a bandwidth of 0.5 nm. The concentration of all VEPs tested was 50 μmol L$^{-1}$, and the measurements were performed in water, mixture of water and trifluoroethanol (TFE) in a 3:2 ratio, mixture of water and methanol (MeOH) in a 1:1 ratio, and sodium dodecyl sulfate (SDS) in water at 10 mmol L$^{-1}$, with respective baselines recorded prior to measurement. A Fourier transform filter was applied to minimize background effects. Helical fraction values were calculated using the single spectra analysis tool on the server BeStSel[39]. Ternary plots were created in https://www.ternaryplot.com/ and subsequently edited.

## Outer membrane permeabilization assays

*N*-phenyl-1-napthylamine (NPN) uptake assay was used to evaluate the ability of the peptides to permeabilize the bacterial outer membrane. Inocula of *P. aeruginosa* PAO1 were grown to an OD at 600 nm of 0.4 mL$^{-1}$, centrifuged (9391 × *g* at 4 °C for 10 min), washed, and resuspended in 5 mmol L$^{-1}$ HEPES buffer (pH 7.4) containing 5 mmol L$^{-1}$ glucose. The bacterial suspension was added to a white 96-well plate (100-μL per well) together with 4 μL of NPN at 0.5 mmol L$^{-1}$.

Consequently, peptides diluted in water were added at their MIC to each well, and the fluorescence was measured at $\lambda_{ex} = 350$ nm and $\lambda_{em} = 420$ nm over time for 45 min. The relative fluorescence was calculated using the untreated control (buffer + bacteria + fluorescent dye) and polymyxin B (positive control) as baselines, and the following equation was applied to reflect % of difference between the baselines and the sample (Eq. 3):

$$\% \, difference = \frac{100 * (fluorescence_{sample} - fluorescence_{untreated \, control})}{fluorescence_{untreated \, control}} \quad (3)$$

## Cytoplasmic membrane depolarization assays

The cytoplasmic membrane depolarization assay was performed using the membrane potential-sensitive dye 3,3'-dipropylthiadicarbocyanine iodide (DiSC$_3$-5). *P. aeruginosa* PAO1 and *S. aureus* ATCC 12600 in the mid-logarithmic phase were washed and resuspended at 0.05 OD mL$^{-1}$ (optical value at 600 nm) in HEPES buffer (pH 7.2) containing 20 mmol L$^{-1}$ glucose and 0.1 mol L$^{-1}$ KCl. DiSC$_3$-5 at 20 μmol L$^{-1}$ was added to the bacterial suspension (100-μL per well) for 15 min to stabilize the fluorescence which indicates the incorporation of the dye into the bacterial membrane, and then the peptides were mixed 1:1 with the bacteria to a final concentration corresponding to their MIC values. Membrane depolarization was then followed by reading changes in the fluorescence ($\lambda_{ex} = 622$ nm, $\lambda_{em} = 670$ nm) over time for 60 min. The relative fluorescence was calculated using the untreated control (buffer + bacteria + fluorescent dye) and polymyxin B (positive control) as baselines and Eq. 3 was applied to reflect % of difference between the baselines and the sample.

## Eukaryotic cell culture

HEK293T cells were obtained from the American Type Culture Collection (CRL-3216). The cells were cultured in high-glucose Dulbecco's modified Eagle's medium supplemented with 1% penicillin and streptomycin (antibiotics) and 10% fetal bovine serum and grown at 37 °C in a humidified atmosphere containing 5% CO$_2$.

## Cytotoxicity assays

One day before the experiment, an aliquot of 100 μL of the cells at 50,000 cells per mL was seeded into each well of the cell-treated 96-well plates used in the experiment (that is, 5000 cells per well). The attached HEK293T cells were then exposed to increasing concentrations of the peptides (8–128 μmol L$^{-1}$) for 24 h. After the incubation period, we performed the 3-(4,5-dimethylthiazol-2-yl)-2,5-diphenyltetrazolium bromide tetrazolium reduction assay (MTT assay). The MTT reagent was dissolved at 0.5 mg mL$^{-1}$ in medium without phenol red and was used to replace cell culture supernatants containing the peptides (100 μL per well), and the samples were incubated for 4 h at 37 °C in a humidified atmosphere containing 5% CO$_2$ yielding the insoluble formazan salt. The resulting salts were then resuspended in hydrochloric acid (0.04 mol L$^{-1}$) in anhydrous isopropanol and quantified by spectrophotometric measurements of absorbance at 570 nm. All assays were done as three biological replicates.

## Hemolysis assays

To assess the release of hemoglobin from human erythrocytes upon treatment of each of the peptides, human red blood cells (RBCs) were obtained from ZenBio (bloodtype A-) heparin anti-coagulated blood. RBCs were washed with PBS (pH 7.4) three times by centrifugation at 800×*g* for 10 min. Aliquots of 250-fold diluted cells (75 μL) were mixed with peptide solution (1–128 μmol L$^{-1}$; 75 μL), and the mixture was incubated for 1 h at room temperature. After the incubation, the plate was centrifuged at 1300×*g* for 10 min to precipitate cells and debris, and an aliquot of 100 μL of supernatant from each well was transferred to a new flat-bottom 96-well plate for absorbance reading (405 nm)

using an automatic plate reader. The percentage of hemolysis was defined by comparison with negative control (samples containing PBS) and positive control [samples containing 1% (v/v) SDS in PBS solution] (Eq. 4).

$$Hemolysis\,(\%) = \frac{100 \times (Absorbance_{peptide} - Absorbance_{negative\,control})}{(Absorbance_{positive\,control} - Absorbance_{negative\,control})}$$

(4)

#### Skin abscess infection mouse model

The back of 6-week-old female CD-1 mice (Charles River stock number: 18679700-022) under anesthesia (isoflurane) were shaved and injured with a superficial linear skin abrasion made with a needle. An aliquot of *A. baumannii* ATCC 19606 ($9.6 \times 10^5$ CFU mL$^{-1}$; 20 µL) previously grown in LB medium until OD (optical value at 600 nm) 0.5 and then washed twice with sterile PBS (pH 7.4, 9,391×*g* for 2 min) was added to the scratched area. Peptides diluted in sterile water at MIC value were administered to the wound area 2 h after the infection. Two- and four-days post-infection animals were euthanized, and the scarified skin was excised, homogenized using a bead beater (25 Hz for 20 min), tenfold serially diluted, and plated on McConkey agar plates for CFU quantification. The experiments were performed using six mice per group ($n = 6$). Mice were single-housed to avoid cross-contamination and maintained under a 12-h light/dark cycle at 22 °C with controlled humidity at 50%. The skin abscess infection mouse model was revised and approved by the University Laboratory Animal Resources (ULAR) from the University of Pennsylvania (Protocol 806763).

#### Quantification and statistical analysis

**Reproducibility of the experimental assays.** Unless otherwise stated, all assays were performed in three independent biological replicates as indicated in each figure legend and experimental models and methods details section. The values obtained for cytotoxic activity were estimated by nonlinear regression based on the screen of peptides in a gradient of concentrations and represent the cytotoxic concentration values needed to lyse and kill 50% of the cells present in the experiment. In the skin abscess mouse model, we used six mice per group following established protocols approved by the University Laboratory of Animal Resources (ULAR) of the University of Pennsylvania.

**Statistical tests.** In the mouse experiments, all the raw data underwent $\log_{10}$ transformation and the statistical significance was determined using one-way ANOVA followed by Dunnett's test. All the *P* values are shown for each of the groups, all groups were compared to the untreated control group. CC$_{50}$ and HC$_{50}$ values were derived from dose-response curves obtained via nonlinear regression analysis, representing the concentrations required to kill 50% of the cells in the experiment.

#### Statistical analysis

All calculations and statistical analyses of the experimental data were conducted using GraphPad Prism v.10.0.2. Statistical significance between different groups was calculated using the tests indicated in each figure legend. No statistical methods were used to predetermine sample size.

#### Reporting summary

Further information on research design is available in the Nature Portfolio Reporting Summary linked to this article.

#### Data availability

The main data generated in this study are available within the paper. The datasets Supplementary Data 1, 2 have been deposited in Mendeley Data under https://doi.org/10.17632/9m4g52grhj.1. All data generated in this study, are available from the corresponding author on reasonable request. Source data are provided as a Source Data file. Source data are provided with this paper.

#### Code availability

APEX is available at GitLab: https://gitlab.com/machine-biology-group-public/apex.

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

## Acknowledgements

Cesar de la Fuente-Nunez holds a Presidential Professorship at the University of Pennsylvania and acknowledges funding from the Procter & Gamble Company, United Therapeutics, a BBRF Young Investigator Grant, the Nemirovsky Prize, Penn Health-Tech Accelerator Award, Defense Threat Reduction Agency grants HDTRA11810041 and HDTRA1-23-1-0001, and the Dean's Innovation Fund from the Perelman School of Medicine at the University of Pennsylvania. Research reported in this publication was supported by the Langer Prize (AIChE Foundation), the NIH R35GM138201, and DTRA HDTRA1-21-1-0014. We thank Dr. Mark Goulian for kindly donating the following strains: *Escherichia coli* AIC221 [*Escherichia coli* MG1655 phnE_2::FRT (control strain for AIC222)] and *Escherichia coli* AIC222 [*Escherichia coli* MG1655 pmrA53 phnE_2::FRT (polymyxin resistant)]. We thank de la Fuente Lab members for insightful discussions. Figures created with BioRender.com are attributed as such. Molecules were rendered using the PyMOL Molecular Graphics System, Version 3.1.1 Schrödinger, LLC.

## Author contributions

Conceptualization, methodology, writing—original draft, and writing—review & editing: M.D.T.T., C.G. and C.d.l.F.N. Experimental investigation: M.D.T.T. and S.L. Computational investigation: C.G. Visualization: C.G. and M.D.T.T. Funding acquisition and supervision: C.d.l.F.N. Formal analysis: M.D.T.T. and C.G.

## Competing interests

Cesar de la Fuente-Nunez is a co-founder and scientific advisor to Peptaris, Inc., provides consulting services to Invaio Sciences and is a member of the Scientific Advisory Boards of Nowture S.L. Peptidus, European Biotech Venture Builder, the Peptide Drug Hunting Consortium (PDHC), ePhective Therapeutics, Inc., and Phare Bio. The de la Fuente Lab has received research funding or in-kind donations from United Therapeutics, Strata Manufacturing PJSC, and Procter & Gamble, none of which were used in support of this work. M.D.T.T. is a co-founder

and scientific advisor to Peptaris, Inc. The remaining authors declare no competing interests. An invention disclosure associated with this work has been filed.
