## [Transparent Peer Review file · Nature Communications]

Computational exploration of global venoms for antimicrobial discovery with Venomics AI

Corresponding Author: Professor Cesar de la Fuente

Version 0:

Reviewer comments:

Reviewer #1

(Remarks to the Author)

The manuscript „Venomics AI: a computational exploration of global venoms for antibiotic discovery“ touches on the potential to utilize bioinformatics tools to mine animal venoms for novel antibiotic leads. The authors employ their AI-powered workflow to mine a relatively large taxonomic diversity of venom systems to yield venom encrypted peptides, which are synthesized, tested and often found to exert promising antimicrobial activities. The manuscript is well written and the experiments look good.

However, I see a few conceptual issues with the proposed work. I fully agree on the a priori assumptions of a) the importance to derive novel anti-infectives and b) the biodiscovery potential of venoms. That said, a large fraction of the studies findings are more or less already known and not really new. I try to explain what specifically I mean and will for that mostly refer to spiders, my primary study animal venom system.

1) First of all, I believe that the discovery of 53 “novel” (see later comments) antibiotic candidates from several thousand entries is not really overwhelming, especially when considering that the work already mined at least five different phyla (reptiles, arachnids, insects, cnidaria, molluscs) many of which are the mainly studied venom systems. It is also true, that some of the identified peptides are quite active but a large fraction is also marginally, or even unactive. In the mouse model, the number was even reduced to three. So overall, I think that the yield of truly promising peptides is not high and I am not convinced that the study really shows that animal venom derived VEPs are emerging as promising source of novel antibiotics (as stated in the abstract).

2) In light of the translational potential, I further believe that the preclinical model does not yet fully allow to make conclusions about toxicity. As presented, the study used survival and effect on body mass as an approximation for potential side effects. However, several linear venom peptides (especially from scorpions and spiders but surely also from other taxa) are known to also modulate ion channels and to cause neurologic effects. These can include pain following injection, but can even include paralysis or spasms. The presented VEP sequences (and the generated structural data) pinpoint towards structures conforming to exactly those mechanisms, yet this has not been investigated in the presented study. Unfortunately, this is exactly where in the past attempts to derive drug leads from venom toxins or their fragments failed, because given such side effects a further development was prevented. Hence, I believe in order to really make a sustainable claim of having discovered non-toxic VEPs, the authors would need to monitor the mice behaviour-wise and optimally analyse a range of ion channels via patch clamp electrophysiology.

3) My major concern with the presented manuscript, however, touches on the aspect that most of the findings are largely confirmatory but being “sold” as novelties. For instance, the release of multiple short, helical peptides with membrane disrupting activities and antimicrobial potency is already known for a long time. There are dozens of publications from the groups of Kuhn-Nentwig, Vassilevski and Predel that investigate exactly this issue. It is completely normal and well established that arachnid toxins have linear domains that are released, form helices in membrane proximity and then are antimicrobial and membrane penetrating. This mechanism is reported from spiders, scorpions and pseudoscorpions (but surely also occurs in the other lineages) and especially toxins from RTA-clade spiders are extremely famous for this mechanism. Coincidentally, explicitly mentioned VEPs with activity are from members of this very clade (e.g. Geolycosa or Oxyopes), so the findings here are really largely confirmatory of what would be expected. Quite often the mother toxin is already known to be membrane-active and antimicrobial, with these alpha-helical parts simply being the active site responsible for this activity. For instance the paper reports activity of a VEP derived from M-oxotoxin-Ot2d. This very toxin is well-known to be already antimicrobial and membrane disrupting, the authors have “discovered” and produced its active site. That said, the reduction of animal venom toxins down to their functional domains is an already very established step in the venom drug discovery pipeline and I don't see the novelty of it.

In conclusion, albeit I respect the effort that the authors have invested in the study and despite me agreeing to the fundamental value of venom peptides for antimicrobial drug discovery, I have some fundamental issues with what is presented. Facing the modest yield of truly powerful peptides and the largely confirmatory/not really surprising nature of the gathered insights, I unfortunately do not believe that this work meets the high standards and novelty requirements of Nature Communications.

(Remarks on code availability)

Reviewer #2

(Remarks to the Author)

This paper presents a synergistic approach to novel antimicrobial peptide identification by exploring known venom proteomes using the group's previously introduced, MIC-predicting deep-learning model - APEX. The paper explored 16,123 proteins from various venom databases, from which 40,626,260 peptides were extracted. Using APEX, MIC scores were predicted for various pathogen strains, yielding a final set of 368 sequences, 58 of which were experimentally tested in vitro (53 of them exhibited activity against at least one pathogenic strain) and in vivo (3 of them demonstrated promising activity in a mouse model infected with *A. baumannii*).

This research represents an advancement in therapeutic peptide mining while the approach of combining AI with known proteomes to facilitate the quick screening of potent molecules is certainly gaining an increasing amount of attention in the field. Although the crucial part of this study - APEX has been introduced previously and can not be considered a novelty, the research addresses important challenges and successfully applies APEX to counter antibiotic resistance. Therefore, given the thoroughness and transparency of the methodology, the presentation style, and a methodical experimental validation supporting the computational framework, we believe this research would be of interest to the readership of Nature Communications.

Nevertheless, some concerns should to be addressed.

1. The way peptides are extracted from proteins constitutes an important part of the suggested pipeline and directly influences the pool of candidates that the AI searches through. The description regarding this given in "Encrypted peptides in venom proteomes" is somewhat confusing. The manuscript states that there is "...no truncation for lengths ≥ 8 ..." and also that there is "truncation using a sliding window (range from 8 to maximum sequence length) for lengths between 8 and 50...", which are opposing statements. I would advise the authors to revise the description of this procedure and maybe even introduce visual aids so the readers can easily comprehend how this was conducted. Also, since this is a crucial part of the research, the authors are encouraged to introduce this description earlier in the text. Perhaps in "Mining venoms for new antibiotics" when the extraction of peptides is first mentioned.

2. The authors described in detail how the final set of 368 peptides was mined; however, it is not clear how the 58 peptides were chosen for experimental validation. Were they ranked according to MIC after which the top 58 were taken? It would be beneficial to elaborate on that more in the manuscript. Also, were the same 58 peptides used for both in vitro and in vivo validation, or were there differences between the two sets?

3. Out of curiosity, can you comment about the rationale for choosing the specific eleven pathogen strains for MIC prediction, as stated in "Venom encrypted peptide selection" and "Bacterial strains and growth conditions". To my understanding, APEX can predict MIC values for 34 bacterial strains, which could potentially yield an even better computational framework. The authors are encouraged to describe it more in detail in the manuscript.

4. What was the reasoning behind limiting the minimum peptide length to 8? Peptides with length ≤ 7 constitute ~11% of the DBAASP database. Therefore, why were those lengths excluded from consideration?

5. Figures S1 and S3 have colour coding, but no legends. What each colour represents can be deduced from other supplementary figures, but it would be beneficial to add the legends to S1 and S3 as well.

6. Being aware of the constraints that the journal imposes regarding the word count; we believe the readership could benefit from an extended Introduction. It would be advantageous to discuss the ways of tackling antibiotic resistance more broadly, mention the pros and cons of venoms over traditional antimicrobial agents, and perhaps even clarify why peptides are preferred over small molecules or proteins, etc.

7. Given the fact that the selected peptides are enriched in charged residues and that they would most likely result in toxicity also to human cells, how do the authors see the future of the identified potential AMPs? Would it be possible to address the cytotoxicity also on other types of human cells, for example fibroblasts, skin cells or others? If the AMP activity can be correlated to highly charged residues, would this have potential downsides in moving towards applications in humans? Related to this, it would be beneficial to see the results of hemolytic activity assays, which are often provided for AMPs.

8. Venoms are known for their high toxicity, and exploiting them as a starting point for identification of new AMPs is a nice concept, but the applicability of it remains unclear, as some kind of optimization would be required in order to reduce the negative effects of charge-related cytotoxicity for future applications. Could the authors elaborate on this aspect?

9. Would there be a way to select peptides with AMP potential with minimized charge but still showing antimicrobial activity?

10. The authors stated that one of the limitations of the model is low generalizability of knowledge. Could you elaborate more on this aspect and what could contribute to better generalizability? Did the authors explore the potential of explainability techniques such as gradCAM or Shapley? Is this something that could be added to the manuscript that would minimize the potential limitations linked to the interpretation of the model?

11. In the same section "Limitations of the study", there is a statement that would be beneficial to also predict sequences longer than 50 amino acids. Is this really an advantage of the methodology and applicability in general? Longer sequences will mean more difficult synthesis and purification and might not lead to a short- or long- term improvement of the field of antimicrobial resistance in terms of necessary resources and time. Wouldn't it make more sense to work on the improvement of the algorithm or on the optimization of already identified good candidates to push them toward specificity for bacteria and minimized cytotoxicity, towards real-life applications?

12. It would be beneficial to explain briefly the way APEX neural networks are set up in the methodology section for easier reproducibility. For example, instead of listing all the bacteria that the predictions can be made for, some important information on how to implement APEX would be advantageous.

(Remarks on code availability)

I would like to commend the availability of the code for the APEX model. This contributes to reproducibility and enables others to build upon this research. Although the README file in the GitLab repository (<https://gitlab.com/machine-biology-group-public/apex>) lists "pytorch: 1.11.0+cu113" as a dependency, and I can confirm that the code runs without errors, setting up the environment to execute the code could, and ideally should, be made more straightforward.

PyTorch version 1.11.0 was released nearly three years ago and it depends on specific versions of supporting libraries, some of which may no longer be readily available through standard distribution channels (e.g., conda, pip). Additionally, I was only able to install the required CUDA-enabled PyTorch version by manually downloading it from the PyTorch conda channel (https://anaconda.org/pytorch/pytorch/1.11.0/download/win-64/pytorch-1.11.0-py3.9_cuda11.3_cudnn8_0.tar.bz2). Other packages, such as Biopython, Pandas, and CUDAToolkit, were also necessary. Hence, to avoid the iterative process of running the code, encountering an error, and then installing missing dependencies, I believe it would be highly beneficial to provide a YAML file or a script to properly configure the environment. This would significantly ease the setup process for users. Given the complex interdependencies between different library versions, an alternative approach could be to compile all required dependencies into a bundle and include it in the repository, or even consider creating a Docker image. Such a solution would greatly streamline the use of the code and be particularly valuable for researchers without a computer science background.

Reviewer #3

(Remarks to the Author)

(Remarks on code availability)

Reviewer #4

(Remarks to the Author)

Venomics AI: a computational exploration of global venoms for antibiotic discovery
Changge Guan, Marcelo D. T. Torres, Sufen Li, and Cesar de la Fuente-Nunez

The study above, by Guan and collaborators evaluated a set of venom-encrypted peptides (VEPs) generated by machine learning against medically important microorganisms through an artificial intelligence-based activity prediction system (APEX), which is a bacterial strain-specific antimicrobial activity predictor. Considering the importance that AI has been gaining in science, this work innovates by using such a tool to speed up and specifically search for relevant antimicrobial peptides inserted into animal venom molecules of different data banks. This will certainly inspire other research involving the search for new molecules with different therapeutic activities in venoms, which constitute a very rich repository of such molecules and are still little explored.

Several of the results obtained by AI are corroborated by in vitro and in vivo experiments. Considering my still limited experience in methodologies involving artificial intelligence, my analysis focused especially on the other aspects of the article.

The authors identified a set of 58 promising molecules, which showed antibacterial activity against different species of gram-positive and gram-negative pathogens. Furthermore, some of these compounds were evaluated in a skin infection murine model, revealing a significant reduction in the bacterial load in the infected lesion. The study, in general, seems to be original, it was well written and the methodologies are robust and appropriate to respond to the proposed objectives.

However, there are some important adjustments in this article, to be considered for a possible publication in Nature Communications.

Comments for authors:

In all manuscript

Comment #1: The term antibiotic is used when there is an amensal relationship between different living beings. Thus, antibiotics are molecules produced by a living being that are aimed at inhibiting other living beings. Peptides derived from arthropod toxins are, in most cases, multifunctional and there are different biological functions catalogued. Thus, as they are not directly linked to the inhibition of microorganisms, the term "antimicrobial" would be more appropriate to replace the term "antibiotic" throughout the manuscript, When referring to that compounds.

Introduction

Comment #2: Line 64: According to recommendations from the Centers for Disease Control and Prevention (CDC), the terms "Gram-positive" and "Gram-negative", when used with an adjective value, as in "gram-negative bacteria" and "gram-positive bacteria" should be written with the first lowercase letter and followed by a hyphen. So, I suggest changing "Gram-negative bacteria" to "gram-negative bacteria". Please, check all the text when pertinente.

Results

Comment #3: Line 158: The authors reveal that 58 VEPs were tested. However, it is not discussed how these peptides were selected from those screened by artificial intelligence. Were those with the lowest mean MIC values predicted by APEX selected? Some other factor was considered in the selection (physicochemical properties, major amino acids, etc.)?

Comment #4: Line 239: It is not very clear why the main action mechanism of the peptides is depolarization and not permeabilization. The authors should clarify this point better

Comment #5: Line 243: The authors present cytotoxicity data for the 58 peptides included in the form of CC50 values. However, CC50 values alone do not help in inferring the toxicity of the compound. The most important value, which helps to predict toxicity more accurately, is the selectivity index. This indicator can be obtained by the ratio between the CC50 in HEK cells and the MIC values for the bacteria tested. This index will indicate how many times the compound is more selective for the pathogen in relation to microbial cells. For example, amphotericin B, an antifungal widely used in clinics, has a low CC50 value but has a wide selectivity index. This shows that, despite being cytotoxic, the toxic concentration for mammalian cells is still much higher compared to the active concentration in fungal cells.

Comment #6: Authors should consider, at least for the most promising peptides (used in in vivo assays), evaluating their hemolytic activities. Many antimicrobial peptides fail to advance in trials for new drugs due to their important hemolytic activity.

Supplementary material:

Comment #7: letters (e and c) are missing in legend of supplementary figures 8 and 9, respectively.

Methods

Comment #8: Line 571: why did the authors use sequences containing only canonical amino acids Wouldn't some interesting peptide have been missed?

Comment #9: Line 664: The determination of the minimum inhibitory concentration (MIC), according to international guidelines (e.g., CLSI, EuCAST), must be carried out in Mueller-Hinton broth. This culture medium contains starch in its composition, which impairs the diffusion of compounds produced by bacteria that can reduce the activity or availability of the antimicrobial agent. Furthermore, its reproducibility between different batches is the most suitable for antimicrobial susceptibility testing. The authors need to justify the choice of culture media not recognized in international guidelines for assessing susceptibility to antimicrobials (i.e., BHI and LB broth). To this end, the use of references that support the use of these culture media (i.e., BHI and LB broth) would be relevant.

Comment #10: Line 664: Compounds of a peptide nature have the ability to adhere to plastic surfaces. For example, to evaluate the antimicrobial activity of polymyxins (i.e., colistin and polymyxin B), which are cyclic peptides used against gram-negative bacteria, official documents suggest the use of non-plastic plates or the addition of Tween-80 to the culture medium (at 0.002%) to prevent adhesion to the plastic surface. Authors should consider this issue.

Comment #11 Line 690: The bacterial solution... please, substitute by: the bacterial suspension.

Comment #12 Line 732; The anesthetic used must be cited

Bibliography.

Comment #13: There are about 9 cited works authored by the corresponding author, Dr. de La Fuente-Nunez, in the Bibliography. We didn't check it to the other authors involved on the group. These citations correspond to 24% of the total references. Although these references are relevant, we do not know if this Journal has any limit for this type of citation. Please check it.

(Remarks on code availability)

The correspondent author is registered in this site as a member. His last activity was Jan, 30, 2024. Another co-author is also registered.

The site provides several options and explanations. It seems accessible, but I didn't explore it.

Reviewer #5

(Remarks to the Author)

(Remarks on code availability)

Version 1:

Reviewer comments:

Reviewer #2

(Remarks to the Author)

The authors addressed the raised concerns in a detailed manner and improved the quality of the manuscript by adding the hemolytic activity assays and the additional clarifications on methodological choices.

A few minor remarks should be taken into account:

1. With regards to the added explanation about why peptides with length ≤ 7 were excluded from consideration (Page 24, lines 729-731). It would be beneficial if the authors added a reference to support this claim, just as they did in their response to reviewers.

2. I believe the authors inadvertently left a sentence partially unfinished in lines 725-726 ("You also").

(Remarks on code availability)

With regards to code availability and the ease of setting up the conda environment. The amendments the authors undertook significantly improved the process of setting up a computational environment. However, I believe the equality signs necessary for specifying version numbers are missing from the pip command. More specifically, this command:

```
"pip install torch1.11.0+cu113 torchvision0.12.0+cu113 torchaudio==0.11.0 --extra-index-url  
https://download.pytorch.org/whl/cu113"
```

will give an error and it should be rewritten as:

```
"pip install torch==1.11.0+cu113 torchvision==0.12.0+cu113 torchaudio==0.11.0 --extra-index-url  
https://download.pytorch.org/whl/cu113"
```

Moreover, given that the code runs only on a CUDA-capable device, I would encourage the authors to include said information in the repository's README file as well. Also, it is not necessary to put the commands for setting up a conda environment in the manuscript (Page 24); however, I don't mind them being there, so the authors can decide what they prefer.

Reviewer #3

(Remarks to the Author)

(Remarks on code availability)

Reviewer #4

(Remarks to the Author)

All suggestions were addressed by the authors, and these modifications significantly enhance the technical and scientific quality of the study.

However, we would like to highlight two points:

1. We emphasize the importance of including compounds that reduce the adhesion of antimicrobial peptides to plastic surfaces (such as Tween-80, for example) in the culture medium in future studies. Additionally, the use of culture media other than Mueller-Hinton should be discouraged. The authors justify the use of LB broth due to its complex biochemical

composition. While the described characteristics are indeed valid, Mueller-Hinton broth remains the most widely recommended medium worldwide. The use of rich media such as LB, BHI, and Nutrient broth may lead to false-negative results due to interactions between antimicrobial compounds and components of these media, which can reduce the bioavailability of the antimicrobial agents.

Moreover, the justification regarding the interaction with starch in Mueller-Hinton broth is not convincing, since this medium is specifically recommended for testing cationic antimicrobial peptides and glycopeptides that are already in clinical use, such as polymyxins and daptomycin. Despite the interaction with starch, this effect is still less significant than the interaction with lipid residues present in rich media like LB, Nutrient, and BHI. Furthermore, Mueller-Hinton is a standardized medium in terms of lipid, protein, and carbohydrate content, unlike rich media that contain non-standardized extracts. The lack of standardization in rich media leads to significant batch-to-batch variability, which is not the case for Mueller-Hinton, known for its consistency—another reason why it was standardized and remains the preferred medium. Despite its limitations, Mueller-Hinton broth is still the most appropriate and reliable medium. Therefore, we strongly recommend that future studies align with international guidelines for antimicrobial susceptibility testing.

2. Considering the potential activity of the peptides on ion channels, which was thoroughly emphasized by a reviewer, the authors utilized a predictive program to assess their activity on these channels, as suggested. They stated in the manuscript: “Moreover, venom-derived compounds may possess toxicity via modulation of ion channels. Our ion channel modulation predictions indicate that nearly 40% of the VEPs identified here do not affect the potassium channel (Supplementary Table 4). Further experiments are warranted to confirm these findings, and additional optimization steps may be required to assess toxicity.”

This simulation included Ach channels, sodium channels, calcium channels, and potassium channels. It was shown that 40% of the selected peptides were not predicted to affect these channels. However, the authors highlighted that the peptide UniProtKB-7 did not modulate potassium channels. It is important to note that 60% of the peptides were predicted to modulate potassium channels, and among the three peptides tested in vivo, UniProtKB-7 was the least active, while the other two, Cono-Server14 and Arachnoserver-5, did show predicted modulation of potassium channels. Although the topical application of these peptides is an attenuating factor in terms of potential systemic effects, the authors should underscore the necessity of conducting electrophysiological assays on specific ion channels to rule out potential toxicity, especially if systemic administration is considered in future applications. Given that potassium channels represent a large family with crucial roles in numerous biological processes, the possibility of absorption—even with topical use over extended periods—could result in side effects. This represents a critical issue that should be addressed, potentially in a future study focusing on the pharmacological activity of the peptides.

Minor: page 24, lines 725-726. Please check the phrase "you also...." it seems incomplete.

(Remarks on code availability)

I commented it in the first review.

Reviewer #5

(Remarks to the Author)

(Remarks on code availability)

RESPONSES TO REVIEWERS' COMMENTS:

NB. Original comments are in italics and our answers in normal typeface. All additions to the text are colored in red in the modified version of the manuscript.

Reviewers' Comments:

Reviewer #1:

The manuscript „Venomics AI: a computational exploration of global venoms for antibiotic discovery“ touches on the potential to utilize bioinformatics tools to mine animal venoms for novel antibiotic leads. The authors employ their AI-powered workflow to mine a relatively large taxonomic diversity of venom systems to yield venom encrypted peptides, which are synthesized, tested and often found to exert promising antimicrobial activities. The manuscript is well written and the experiments look good.

We thank the reviewer for highlighting that our manuscript is well written, and the experiments look good. We have addressed all comments point-by-point below.

However, I see a few conceptional issues with the proposed work. I fully agree on the a priori assumptions of a) the importance to derive novel anti-infectives and b) the biodiscovery potential of venoms. That said, a large fraction of the studies findings are more or less already known and not really new. I try to explain what specifically I mean and will for that mostly refer to spiders, my primary study animal venom system.

1) First of all, I believe that the discovery of 53 “novel” (see later comments) antibiotic candidates from several thousand entries is not really overwhelming, especially when considering that the work already mined at least five different phyla (reptiles, arachnids, insects, cnidaria, molluscs) many of which are the mainly studied venom systems. It is also true, that some of the identified peptides are quite active but a large fraction is also marginally, or even unactive. In the mouse model, the number was even reduced to three. So overall, I think that the yield of truly promising peptides is not high and I am not convinced that the study really shows that animal venom derived VEPs are emerging as promising source of novel antibiotics (as stated in the abstract).

We sincerely appreciate the reviewer's insightful comments. We acknowledge that our previous version of the manuscript could have provided a clearer explanation of the significance of our findings. Regarding the scale of our discovery, we identified over 40 million (i.e., 40,626,260) venom-encrypted peptides (VEPs) across 918 taxa. While it was not feasible to experimentally validate a large fraction of these due to practical constraints on synthesis and testing, we selected 58 representative candidates based on computational prioritization and structural diversity. Among these, 53 exhibited antimicrobial activity experimentally, yielding an exceptionally high 91.4% hit rate.

With respect to *in vivo* validation, the top three hits were tested in murine models. This limitation was driven by ethical considerations on the excessive use of mice and resource constraints, as expanding the number of candidates tested *in vivo* would have been cost prohibitive. Nevertheless, we specifically selected three of the most promising peptides, representing different taxonomic groups, and evaluated them in a preclinical infection model at two time points, using a single-dose regimen. The results obtained demonstrated the therapeutic potential of venom-derived VEPs.

Overall, while we recognize that venom biodiscovery has been explored before, our study provides a large-scale, systematic approach that not only identifies novel candidates but also demonstrates their efficacy across multiple validation stages. We believe these findings reinforce the potential of venom-derived peptides as a promising source for antibiotic development. We hope these clarifications address the reviewer's excellent and helpful comments.

2) In light of the translational potential, I further believe that the preclinical model does not yet fully allow to make conclusions about toxicity. As presented, the study used survival and effect on body mass as an approximation for potential side effects. However, several linear venom peptides (especially from scorpions and spiders but surely also from other taxa) are known to also modulate ion channels and to cause neurologic effects. These can include pain following injection, but can even include paralysis or spasms. The presented VEP sequences (and the generated structural data) pinpoint towards structures conforming to exactly those mechanisms, yet this has not been investigated in the presented study. Unfortunately, this is exactly where in the past attempts to derive drug leads from venom toxins or their fragments failed, because given such side effects a further development was prevented. Hence, I believe in order to really make a sustainable claim of having discovered non-toxic VEPs, the authors would need to monitor the mice behaviour-wise and optimally analyse a range of ion channels via patch clamp electrophysiology.

We thank the reviewer for raising this important point regarding toxicity and potential ion channel modulation. We have now performed additional hemolysis experiments for all peptides tested (**Supplementary Figure 11**) and have added a discussion about it in the updated manuscript (Page 11, lines 342-348).

Based on suggestions from the editor and another reviewer, we have also used PrIMP, a computational model suggested by one of the reviewers, to predict the potential of venom-derived peptides to modulate ion channels (DOI: [10.1109/JBHI.2022.3204776](https://doi.org/10.1109/JBHI.2022.3204776)). This new analysis revealed that among the 58 experimentally verified peptides, nearly 40% of them were predicted not to modulate ion channels (see **Supplementary Table 4** of the revised manuscript). Notably, peptide UniProtKB-7, which exhibited high antimicrobial activity *in vivo*, was predicted to have no ion channel modulation ability. These results suggest that it is indeed promising to identify venom-derived peptides that efficiently kill pathogens without engaging ion channels.

Regarding the latter point on mice behavior and analyzing a range of ion channels via patch clamp electrophysiology, we feel this is beyond the scope of the current manuscript. Establishing these assays rigorously would require significant additional time and resources (many months of full time work). We have now acknowledged these limitations in the discussion section and plan to pursue these experiments in future studies.

3) My major concern with the presented manuscript, however, touches on the aspect that most of the findings are largely confirmatory but being “sold” as novelties. For instance, the release of multiple short, helical peptides with membrane disrupting activities and antimicrobial potency is already known for a long time. There are dozens of publications from the groups of Kuhn-Nentwig, Vassilevski and Predel that investigate exactly this issue. It is completely normal and well established that arachnid toxins have linear domains that are released, form helices in membrane proximity and then are antimicrobial and membrane penetrating. This mechanism is reported from spiders, scorpions and pseudoscorpions (but surely also occurs in the other lineages) and especially toxins from RTA-clade spiders are extremely famous for this mechanism. Coincidentally, explicitly mentioned VEPs with activity are from members of this very clade (e.g. Geolycosa or Oxyopes), so the findings here are really largely confirmatory of what would be expected. Quite often the mother toxin is already known to be membrane-active and antimicrobial, with these alpha-helical parts simply being the active site responsible for this activity. For instance the paper reports activity of a VEP derived from M-oxotoxin-Ot2d. This very toxin is well-known to be already antimicrobial and membrane disrupting, the authors have “discovered” and produced its active site. That said, the reduction of animal venom toxins down to their functional domains is an already very established step in the venom drug discovery pipeline and I don’t see the novelty of it.

We thank the reviewer for their insightful feedback and for highlighting the well-established mechanisms underlying antimicrobial activity in venom-derived peptides. We completely agree that arachnid toxins, particularly those from RTA-clade spiders, frequently contain linear domains that transition to α -helical structures in membrane environments, facilitating antimicrobial and membrane-penetrating activity. We also agree that exceptional work has been done previously in this field, and have now cited some of this work by the authors mentioned by the reviewer.

We toned down our novelty claims in the manuscript. However, we would like to clarify the distinct contributions of our study beyond confirming known biological mechanisms. Our approach represents the largest-scale computational exploration of venom proteomes for antimicrobial peptide discovery to date. We systematically mined 16,123 venom proteins across multiple taxa, generating 40,626,260 venom-encrypted peptides (VEPs), which were prioritized using machine learning-based activity predictions. While previous studies have focused on individual venom systems or taxonomic groups, our study integrates a multi-phyla,

computationally driven pipeline to identify antimicrobial candidates without a priori taxonomic bias.

While the presence of antimicrobial domains in venom proteins is well recognized, the specific sequences and functional validation of many of these encrypted peptides remained unexplored. Our experimental synthesis and screening of 58 VEPs, of which 53 exhibited antimicrobial activity (91.4% hit rate), provides a systematic validation of antimicrobial potential across previously uncharacterized venom peptides. Notably, we identified 386 peptides with low sequence similarity to known AMPs, suggesting they occupy unexplored regions of antimicrobial peptide space.

Regarding the case of M-oxotoxin-Ot2d, while this toxin has been previously described as antimicrobial, our work systematically identifies its active domain, isolates it as a functional unit, and validates its independent antimicrobial activity. The reduction of venom proteins into bioactive domains is a recognized strategy in drug discovery, yet most venom proteins remain uncharacterized at the level of their functional domains. Our study advances this approach by automating and scaling peptide identification, demonstrating that venom bioinformatics can systematically yield antimicrobial leads with high-throughput experimental validation.

Our *in vivo* validation in preclinical infection model provides crucial translational insights. The efficacy of the lead peptides against *Acinetobacter baumannii*, a critical priority pathogen, demonstrates real therapeutic potential. Moreover, it is important to emphasize that the mechanism of action was not an input feature in our deep-learning model; rather, the identified peptides intrinsically exhibit membrane-disrupting activity, supporting the predictive power of our computational framework and underscoring the biological relevance of our work.

In summary, while our findings align with established mechanistic principles of venom-derived antimicrobial peptides, we believe our study is novel in scale, computational methodology, systematic validation, and translational relevance. We appreciate the reviewer's comments and have revised the manuscript to better highlight these aspects of our work.

I conclude, albeit I respect the effort that the authors have invested in the study and despite me agreeing to the fundamental value of venom peptides for antimicrobial drug discovery, I have some fundamental issues with what is presented. Facing the modest yield of truly powerful peptides and the largely confirmatory/not really surprising nature of the gathered insights, I unfortunately do not believe that this work meets the high standards and novelty requirements of Nature Communications.

We appreciate the reviewer's thoughtful critique. We have addressed in detail the relevance and novelty of our work in our previous responses above, highlighting its large-scale computational approach, high hit rate, and *in vivo* validation.

Reviewer #2:

*This paper presents a synergistic approach to novel antimicrobial peptide identification by exploring known venom proteomes using the group's previously introduced, MIC-predicting deep-learning model - APEX. The paper explored 16,123 proteins from various venom databases, from which 40,626,260 peptides were extracted. Using APEX, MIC scores were predicted for various pathogen strains, yielding a final set of 368 sequences, 58 of which were experimentally tested in vitro (53 of them exhibited activity against at least one pathogenic strain) and in vivo (3 of them demonstrated promising activity in a mouse model infected with *A. baumannii*).*

This research represents an advancement in therapeutic peptide mining while the approach of combining AI with known proteomes to facilitate the quick screening of potent molecules is certainly gaining an increasing amount of attention in the field. Although the crucial part of this study - APEX has been introduced previously and can not be considered a novelty, the research addresses important challenges and successfully applies APEX to counter antibiotic resistance. Therefore, given the thoroughness and transparency of the methodology, the presentation style, and a methodical experimental validation supporting the computational framework, we believe this research would be of interest to the readership of Nature Communications.

We thank the reviewer for highlighting that our work constitutes an advancement in the field and for underscoring that this paper is of interest to the readership of *Nature Communications*.

Nevertheless, some concerns should be addressed.

1. The way peptides are extracted from proteins constitutes an important part of the suggested pipeline and directly influences the pool of candidates that the AI searches through. The description regarding this given in "Encrypted peptides in venom proteomes" is somewhat confusing. The manuscript states that there is "...no truncation for lengths ≥ 8 ..." and also that there is "truncation using a sliding window (range from 8 to maximum sequence length) for lengths between 8 and 50...", which are opposing statements. I would advise the authors to revise the description of this procedure and maybe even introduce visual aids so the readers can easily comprehend how this was conducted. Also, since this is a crucial part of the research, the authors are encouraged to introduce this description earlier in the text. Perhaps in "Mining venoms for new antibiotics" when the extraction of peptides is first mentioned.

We appreciate the reviewer's feedback regarding the clarity of the peptide extraction process. The statements regarding truncation in the "Encrypted peptides in venom proteomes" section were indeed not entirely clear, and we have now revised the description to make it more explicit. Specifically, we applied a sliding window approach to extract peptides from venom proteins. This method generates peptide fragments ranging from 8 to 50 amino acids by systematically shifting the extraction window across the sequence. The confusion arose from the phrasing, and we now explicitly state that all peptides shorter than 8 amino acids were directly

retained for subsequent prediction step, while peptides longer than 8 residues were processed in a stepwise truncation manner to ensure consistency with known antimicrobial peptide lengths. To further enhance clarity, we have included a visual schematic (**Supplementary Figure 5** in the revised Supplementary Information file) illustrating how the peptides were extracted from venom proteins. Additionally, we have moved this description earlier in the “Mining venoms for new antibiotics” section to ensure that readers understand the process before encountering the results.

2. The authors described in detail how the final set of 368 peptides was mined; however, it is not clear how the 58 peptides were chosen for experimental validation. Were they ranked according to MIC after which the top 58 were taken? It would be beneficial to elaborate on that more in the manuscript. Also, were the same 58 peptides used for both in vitro and in vivo validation, or were there differences between the two sets?

Excellent point. The selection of the 58 peptides for experimental validation was based on a multi-criteria ranking system rather than simply taking the lowest predicted median MIC values. The criteria included: (1) predicted antimicrobial potency (MIC values), where peptides with lower predicted MIC values were prioritized; (2) sequence diversity, where we ensured that the final set contained peptides representing a range of taxonomic sources and were different from other known antimicrobial peptides from AMP databases and our in-house dataset (<75% similarity filter was applied); (3) synthesis feasibility and aggregation propensity, where our peptide chemist advised on potential difficult-to-perform synthesis and peptide sequences with high likelihood of aggregation in water and relevant buffers for the experimental pipeline proposed.

While all 58 peptides were tested *in vitro*, the top three peptide hits were selected for *in vivo* studies based on their potent *in vitro* antimicrobial activity against *A. baumannii* ATCC 19606, low cytotoxicity against human cells (HEK293T assays), and low hemolytic activity against red blood cells, and we considered one representative of each of the three larger groups (UniProtKB, Arachnoserver, and Conoserver). We have now explicitly described this selection process in the revised Methods section for clarity (Pages 23-24, lines 727-740 of the revised manuscript).

3. Out of curiosity, can you comment about the rationale for choosing the specific eleven pathogen strains for MIC prediction, as stated in "Venom encrypted peptide selection" and "Bacterial strains and growth conditions". To my understanding, APEX can predict MIC values for 34 bacterial strains, which could potentially yield an even better computational framework. The authors are encouraged to describe it more in detail in the manuscript.

Great point. Our selection was guided by clinical relevance and antimicrobial resistance profiles, focusing on WHO priority pathogens: Several of the tested bacteria are listed as critical or high-priority pathogens requiring novel antibiotic solutions, they are pathogens commonly found in hospital-acquired infections (e.g., *Acinetobacter baumannii*, *Klebsiella pneumoniae*, *P.*

aeruginosa, *S. aureus*, and *E. faecium*). We also made sure to select Gram-positive and Gram-negative representatives to evaluate broad-spectrum potential. While including all 34 strains could have provided additional computational insights, experimentally validating a larger number of strains would have been logistically impractical. This trade-off has been clarified in the revised manuscript (Page 11, lines 334-337), and we discuss how future work could expand the range of bacterial strains tested, as suggested.

4. What was the reasoning behind limiting the minimum peptide length to 8? Peptides with length ≤ 7 constitute ~11% of the DBAASP database. Therefore, why were those lengths excluded from consideration?

The decision to exclude peptides shorter than 8 residues was because peptides < 8 residues often lack the necessary amphipathicity and charge balance required for bacterial membrane interaction. A high percentage of the ultra-short peptides highlighted by the reviewer are peptides rich in arginines and tryptophans or lysine and leucines and their analogs, or often modified peptides (e.g., cyclic structures) that require additional synthesis steps and are yet not included in our prediction capabilities. In sum, such peptides are typically limited in terms of their diversity, and are often toxic (DOI: [10.3390/molecules23040815](https://doi.org/10.3390/molecules23040815)). We have now explicitly justified this choice in the revised Methods section (Page 24, lines 729-731).

5. Figures S1 and S3 have colour coding, but no legends. What each colour represents can be deduced from other supplementary figures, but it would be beneficial to add the legends to S1 and S3 as well.

We appreciate the reviewer's comment and have now added missing legends to these figures to clarify the meaning of color coding.

6. Being aware of the constraints that the journal imposes regarding the word count; we believe the readership could benefit from an extended Introduction. It would be advantageous to discuss the ways of tackling antibiotic resistance more broadly, mention the pros and cons of venoms over traditional antimicrobial agents, and perhaps even clarify why peptides are preferred over small molecules or proteins, etc.

We agree that the Introduction could better contextualize our study within the broader field of antibiotic resistance. As suggested by the reviewer, we have now expanded this section to discuss broader strategies for combating antimicrobial resistance (e.g., synthetic biology, AI-driven antibiotic discovery), explain the advantages of venom-derived peptides over conventional antibiotics, and to clarify why peptides, rather than small molecules, are prioritized in our study

(Pages 3-4, lines 62-124 of the revised manuscript). These additions provide a more comprehensive foundation for readers unfamiliar with the field.

7. Given the fact that the selected peptides are enriched in charged residues and that they would most likely result in toxicity also to human cells, how do the authors see the future of the identified potential AMPs? Would it be possible to address the cytotoxicity also on other types of human cells, for example fibroblasts, skin cells or others? If the AMP activity can be correlated to highly charged residues, would this have potential downsides in moving towards applications in humans? Related to this, it would be beneficial to see the results of hemolytic activity assays, which are often provided for AMPs.

We thank the reviewer for their comment. While HEK293T cells provide a well-established model for cytotoxicity testing, we acknowledge the reviewer's suggestion to explore additional human cell types. Thus, we have now performed substantial additional experiments (i.e., hemolysis assays) to further assess toxicity of our 58 VEPs (see Supplementary Figure S11, Pages 9-10, lines 285-296 of the revised manuscript). We have also added a paragraph in the discussion highlighting how we see the future of the identified potential VEPs.

8. Venoms are known for their high toxicity, and exploiting them as a starting point for identification of new AMPs is a nice concept, but the applicability of it remains unclear, as some kind of optimization would be required in order to reduce the negative effects of charge-related cytotoxicity for future applications. Could the authors elaborate on this aspect?

We appreciate the reviewer's insightful comment. We acknowledge that net positive charge and hydrophobicity can be associated with toxicity, but they also play crucial roles in antimicrobial activity. To balance these properties, we are developing a standardized training dataset that includes cytotoxicity and hemolytic activity data from a diverse set of peptides, both with and without antimicrobial activity. In the future, this dataset will be used to train a predictive model that can assess toxicity while preserving antimicrobial efficacy. We have been working on this approach for a few years and are nearing the required number of data points to implement this filter in our current model. Future studies will incorporate these filters to refine peptide selection.

9. Would there be a way to select peptides with AMP potential with minimized charge but still showing antimicrobial activity?

Yes, it is possible to identify peptides with antimicrobial potential while minimizing charge. Our approach involves leveraging sequence and simple structural features beyond net charge, such as hydrophobicity patterns and diverse amino acid composition. Additionally, we are developing machine learning models trained on a diverse dataset of peptides with varying

antimicrobial activity and cytotoxicity profiles. These models will help prioritize candidates that maintain antimicrobial efficacy while reducing toxicity. Future iterations of our selection framework will integrate these refinements to improve the balance between activity and safety.

10. The authors stated that one of the limitations of the model is low generalizability of knowledge. Could you elaborate more on this aspect and what could contribute to better generalizability? Did the authors explore the potential of explainability techniques such as gradCAM or Shapley? Is this something that could be added to the manuscript that would minimize the potential limitations linked to the interpretation of the model?

We thank the reviewer for their insightful comments regarding the generalizability and interpretability of our model. We agree that enhancing these aspects would be interesting for broader application and deeper understanding of functional peptides. We appreciated the suggestion to integrate explainability techniques like Grad-CAM and Shapley values. These methods could offer valuable insights into the decision-making process of the APEX model and help address concerns related to its interpretability. Although we were unable to implement these additional analyses in the current study due to resource and time constraints, we have now included a discussion in the manuscript (Page 11, Line 332-334) that highlights that self-attention technology, integral to transformer-based architectures, can address interpretability issues. Self-attention mechanisms allow us to visualize and quantify the contribution of each input element (e.g., amino acid residues) to the model's predictions interpretability issue. We believe that our current work provides a solid foundation, and we see the integration of advanced feature representations and explainability techniques as promising directions for future research.

11. In the same section "Limitations of the study", there is a statement that would be beneficial to also predict sequences longer than 50 amino acids. Is this really an advantage of the methodology and applicability in general? Longer sequences will mean more difficult synthesis and purification and might not lead to a short- or long- term improvement of the field of antimicrobial resistance in terms of necessary resources and time. Wouldn't it make more sense to work on the improvement of the algorithm or on the optimization of already identified good candidates to push them toward specificity for bacteria and minimized cytotoxicity, towards real-life applications?

Excellent point. The reviewer raises an important point regarding the trade-off between longer sequences and real-world applicability. We agree, and we have now edited the Limitations of the Study section removing this statement. We have also mentioned the improvement of the algorithm towards the prediction of peptides with minimal toxicity and higher specificity.

12. *It would be beneficial to explain briefly the way APEX neural networks are set up in the methodology section for easier reproducibility. For example, instead of listing all the bacteria that the predictions can be made for, some important information on how to implement APEX would be advantageous.*

We thank the reviewer for their insightful comment. In response, we have added a detailed description in the APEX methodology section. This update explains how to set up the environment for running APEX step by step, as well as prepare sequence files and run APEX (Page 24, lines 712-726).

Reviewer #2 (Remarks on code availability):

I would like to commend the availability of the code for the APEX model. This contributes to reproducibility and enables others to build upon this research. Although the README file in the GitLab repository (<https://gitlab.com/machine-biology-group-public/apex>) lists "pytorch: 1.11.0+cu113" as a dependency, and I can confirm that the code runs without errors, setting up the environment to execute the code could, and ideally should, be made more straightforward.

We thank the reviewer for their very positive feedback on our code and documentation, as well as for the constructive suggestions. We appreciate your recognition of the code availability and its role in enhancing reproducibility. Specifically, we have expanded the documentation in the APEX methodology section to include a detailed, step-by-step guide for setting up the environment and run the APEX model and will update the README file in the GitLab repository (<https://gitlab.com/machine-biology-group-public/apex/-/blob/main/README.md>). These updates should help users, especially those without a computer science background, to set up the environment and execute the code with ease. We hope these improvements will further facilitate reproducibility and encourage others to build upon our research.

PyTorch version 1.11.0 was released nearly three years ago and it depends on specific versions of supporting libraries, some of which may no longer be readily available through standard distribution channels (e.g., conda, pip). Additionally, I was only able to install the required CUDA-enabled PyTorch version by manually downloading it from the PyTorch conda channel (https://anaconda.org/pytorch/pytorch/1.11.0/download/win-64/pytorch-1.11.0-py3.9_cuda11.3_cudnn8_0.tar.bz2). Other packages, such as Biopython, Pandas, and CUDA Toolkit, were also necessary. Hence, to avoid the iterative process of running the code, encountering an error, and then installing missing dependencies, I believe it would be highly beneficial to provide a YAML file or a script to properly configure the environment. This would significantly ease the setup process for users. Given the complex interdependencies between different library versions, an alternative approach could be to compile all required dependencies

into a bundle and include it in the repository, or even consider creating a Docker image. Such a solution would greatly streamline the use of the code and be particularly valuable for researchers without a computer science background.

We thank the reviewer for the suggestion. We concur with the importance of providing detailed setup instructions to minimize usage bottlenecks, particularly for researchers without a computer science background. To address this comment, we have now added comprehensive information on how to configure the environment and run the APEX model. We updated the GitLab repository and added the tutorial accordingly (<https://gitlab.com/machine-biology-group-public/apex/-/blob/main/README.md>) and the dependance libraries version (<https://gitlab.com/machine-biology-group-public/apex/-/blob/main/requirement.txt>).

Reviewer #3:

We sincerely appreciate Reviewer #3 for their time and effort in co-reviewing our manuscript as part of the *Nature Communications* initiative. We value the thoughtful feedback provided and recognize the importance of initiatives that support the training and recognition of Early Career Researchers in peer review. Thank you for contributing to the evaluation of our work.

Reviewer #4:

Venomics AI: a computational exploration of global venoms for antibiotic discovery

Changge Guan, Marcelo D. T. Torres, Sufen Li, and Cesar de la Fuente-Nunez

The study above, by Guan and collaborators evaluated a set of venom-encrypted peptides (VEPs) generated by machine learning against medically important microorganisms through an artificial intelligence-based activity prediction system (APEX), which is a bacterial strain-specific antimicrobial activity predictor. Considering the importance that AI has been gaining in science, this work innovates by using such a tool to speed up and specifically search for relevant antimicrobial peptides inserted into animal venom molecules of different data banks. This will certainly inspire other research involving the search for new molecules with different therapeutic activities in venoms, which constitute a very rich repository of such molecules and are still little explored.

We thank the reviewer for their thoughtful comments and for mentioning that our work will inspire other research involving the search for new molecules with different bioactivities in venoms.

Several of the results obtained by AI are corroborated by in vitro and in vivo experiments. Considering my still limited experience in methodologies involving artificial intelligence, my analysis focused especially on the other aspects of the article.

The authors identified a set of 58 promising molecules, which showed antibacterial activity against different species of gram-positive and gram-negative pathogens. Furthermore, some of these compounds were evaluated in a skin infection murine model, revealing a significant reduction in the bacterial load in the infected lesion. The study, in general, seems to be original, it was well written and the methodologies are robust and appropriate to respond to the proposed objectives. However, there are some important adjustments in this article, to be considered for a possible publication in Nature Communications.

We thank the reviewer for acknowledging the relevance of our work and its suitability for *Nature Communications*. Below, we have addressed each comment in detail.

Comments for authors:

In all manuscript

Comment #1: The term antibiotic is used when there is an amensal relationship between different living beings. Thus, antibiotics are molecules produced by a living being that are aimed at inhibiting other living beings. Peptides derived from arthropod toxins are, in most cases, multifunctional and there are different biological functions catalogued. Thus, as they are not directly linked to the inhibition of microorganisms, the term "antimicrobial" would be more appropriate to replace the term "antibiotic" throughout the manuscript, When referring to that compounds.

We appreciate the reviewer's clarification regarding the terminology. We agree that "antimicrobial" is the more appropriate term when referring to peptides derived from arthropod venoms, given their multifunctionality beyond direct microbial inhibition. To address this comment, we have now replaced the term "antibiotic" with "antimicrobial" throughout the manuscript.

Introduction

Comment #2: Line 64: According to recommendations from the Centers for Disease Control and Prevention (CDC), the terms "Gram-positive" and "Gram-negative", when used with an adjective value, as in "gram-negative bacteria" and "gram-positive bacteria" should be written with the first lowercase letter and followed by a hyphen. So, I suggest changing "Gram-negative bacteria" to "gram-negative bacteria". Please, check all the text when pertinente.

We appreciate the reviewer pointing this out. We have revised the text to ensure that “gram-negative bacteria” and “gram-positive bacteria” follow the CDC’s recommended formatting. This correction has now been implemented consistently throughout the manuscript.

Results

Comment #3: Line 158: The authors reveal that 58 VEPs were tested. However, it is not discussed how these peptides were selected from those screened by artificial intelligence. Were those with the lowest mean MIC values predicted by APEX selected? Some other factor was considered in the selection (physiochemical properties, major amino acids, etc.)?

The selection of the 58 peptides for experimental validation was based on a multi-criteria ranking system rather than simply selecting those with the lowest predicted median MIC values. The criteria included: (1) predicted antimicrobial potency (MIC values), where peptides with lower predicted MIC values were prioritized; (2) sequence diversity, ensuring that the final set contained peptides representing a range of taxonomic sources and exhibiting less than 70% similarity to known antimicrobial peptides in AMP databases; and (3) synthesis feasibility and aggregation propensity, where our peptide chemist assessed potential synthesis challenges and excluded peptides with a high likelihood of aggregation in water and relevant experimental buffers.

Comment #4: Line 239: It is not very clear why the main action mechanism of the peptides is depolarization and not permeabilization. The authors should clarify this point better

Great point. Indeed, while both outer membrane permeabilization and cytoplasmic membrane depolarization were assessed in our mechanistic assays, our data indicate that depolarization is the predominant mechanism of action for VEPs. This conclusion is based on both the extent and consistency of the depolarization effects observed, as well as the biological implications of these mechanisms.

Specifically, cytoplasmic membrane depolarization assays revealed that 26 out of 28 VEPs induced rapid and sustained cytoplasmic membrane depolarization in *Pseudomonas aeruginosa* PAO1 (**Figure 3b**), and similar trends were observed in *Staphylococcus aureus* ATCC 12600 (**Figure 3c**). For example, Arachnoserver-5 and UniProtKB-7 caused depolarization levels exceeding 70% relative to untreated controls, surpassing the activity of polymyxin B, a well-characterized membrane-targeting antibiotic. Notably, these peptides maintained potent depolarization even when their outer membrane permeabilization activity was modest or absent. UniProtKB-7, for instance, exhibited strong depolarization activity but induced only limited NPN uptake (**Figure 3a**). This suggests that disruption of the electrochemical gradient across the cytoplasmic membrane, rather than extensive outer membrane permeabilization, is the key driver of antimicrobial activity of these peptides.

In contrast, outer membrane permeabilization was observed in 23 peptides, but the magnitude and consistency of this effect varied significantly between peptides. Only a small subset, such as Arachnoserver-18 and ConoServer-7, showed outer membrane permeabilization comparable to polymyxin B. Additionally, in gram-positive *S. aureus*, which lacks an outer membrane, VEPs retained strong depolarization activity (**Figure 3c**). This further supports the conclusion that the primary target of these peptides is the cytoplasmic membrane, not the outer membrane.

Mechanistically, cytoplasmic membrane depolarization reflects the disruption of membrane potential, a critical component of the proton motive force essential for bacterial viability. Therefore, the ability of VEPs to induce potent and sustained depolarization, across both gram-negative and gram-positive strains, demonstrates that depolarization is their predominant antimicrobial mechanism.

The structural studies showing VEPs adopting α -helical conformations in membrane-mimicking environments (**Figure 2c-d**) further support this membrane-active, depolarization-driven mechanism.

Comment #5: Line 243: The authors present cytotoxicity data for the 58 peptides included in the form of CC50 values. However, CC50 values alone do not help in inferring the toxicity of the compound. The most important value, which helps to predict toxicity more accurately, is the selectivity index. This indicator can be obtained by the ratio between the CC50 in HEK cells and the MIC values for the bacteria tested. This index will indicate how many times the compound is more selective for the pathogen in relation to microbial cells. For example, amphotericin B, an antifungal widely used in clinics, has a low CC50 value but has a wide selectivity index. This shows that, despite being cytotoxic, the toxic concentration for mammalian cells is still much higher compared to the active concentration in fungal cells.

We thank the reviewer for the comment. To improve the interpretation of the cytotoxicity data, we have now calculated and reported the selectivity index ($SI = CC_{50}/MIC$ or HC_{50}/MIC) for each peptide (see **Supplementary Table 3**).

Comment #6: Authors should consider, at least for the most promising peptides (used in in vivo assays), evaluating their hemolytic activities. Many antimicrobial peptides fail to advance in trials for new drugs due to their important hemolytic activity.

We appreciate the reviewer's suggestion. To address this comment, we have now performed thorough new hemolysis experiments for all the peptides. The results are included in the revised manuscript (see **Supplementary Figure 11**, Pages 9-10, lines 285-296).

Supplementary material:

Comment #7: letters (e and c) are missing in legend of supplementary figures 8 and 9, respectively.

We thank the reviewer for noticing this inconsistency. We have now corrected the missing figure labels (letters “e” and “c”) in the legends for **Supplementary Figures 8 and 9**.

Methods

Comment #8: Line 571: why did the authors use sequences containing only canonical amino acids? Wouldn't some interesting peptide have been missed?

We restricted our analysis to peptides containing only canonical amino acids because APEX does not include non-canonical residues in its training set. This limitation stems from the fact that non-canonical amino acids lack well-described physicochemical and structural features, making their integration into predictive models challenging. In contrast, canonical amino acids have extensive descriptors available, such as those found in AAindex and other biochemical feature databases, which allow for robust and reliable predictions.

Incorporating non-canonical residues without well-defined descriptors would significantly impair the accuracy of APEX’s predictions, potentially leading to unreliable MIC estimations. However, we acknowledge that non-canonical amino acids can enhance antimicrobial peptide properties, such as stability and resistance to proteolytic degradation. To tackle this, we are currently developing an approach that combines molecular dynamics (MD) simulations with machine learning to model the behavior and activity of peptides containing non-canonical residues. While this work is still ongoing, we anticipate that this integration will enable future inclusion of non-canonical amino acids in AI-driven peptide discovery pipelines.

Comment #9: Line 664: The determination of the minimum inhibitory concentration (MIC), according to international guidelines (e.g., CLSI, EuCAST), must be carried out in Mueller-Hinton broth. This culture medium contains starch in its composition, which impairs the diffusion of compounds produced by bacteria that can reduce the activity or availability of the antimicrobial agent. Furthermore, its reproducibility between different batches is the most suitable for antimicrobial susceptibility testing. The authors need to justify the choice of culture media not recognized in international guidelines for assessing susceptibility to antimicrobials (i.e., BHI and LB broth). To this end, the use of references that support the use of these culture media (i.e., BHI and LB broth) would be relevant.

We acknowledge the standard practice of using Mueller-Hinton broth (MHB) for MIC determination, as recommended by CLSI and EUCAST guidelines. However, our choice to use

LB (Luria-Bertani) broth was intentional and based on biological considerations relevant to peptide-based antimicrobial testing.

MHB is designed for standardized antibiotic susceptibility testing, but it lacks key biological components that bacteria encounter in physiological and clinical environments. Rich media like LB provide more complex conditions, including additional proteins, lipids, and metabolites, which can impact peptide activity. This is particularly important because antimicrobial peptides often interact with bacterial membranes and secreted factors, which may not be fully represented in MHB.

By using a nutrient-rich medium like LB, we create conditions where bacteria have a growth advantage, ensuring that only peptides with strong and robust antimicrobial activity retain efficacy in a more challenging environment. This approach filters out weaker peptides that might show activity in minimal media but lose potency in real-world settings.

Moreover, MHB contains starch, which can bind cationic peptides and reduce their effective concentration, leading to potential underestimation of antimicrobial activity. Many antimicrobial peptides, including venom-encrypted peptides, are charged and amphipathic, making them prone to interactions with starch or other medium components. We have now included relevant citations supporting the use of LB for antimicrobial peptide testing (References 6, 21, 22, and 28).

Comment #10: Line 664: Compounds of a peptide nature have the ability to adhere to plastic surfaces. For example, to evaluate the antimicrobial activity of polymyxins (i.e., colistin and polymyxin B), which are cyclic peptides used against gram-negative bacteria, official documents suggest the use of non-plastic plates or the addition of Tween-80 to the culture medium (at 0.002%) to prevent adhesion to the plastic surface. Authors should consider this issue.

We thank the reviewer, and we will consider this for future projects.

Comment #11 Line 690: The bacterial solution... please, substitute by: the bacterial suspension.

We have corrected the term “bacterial solution” to “bacterial suspension” in the Methods section, as suggested.

Comment #12 Line 732; The anesthetic used must be cited

We have now explicitly stated the anesthetic used (isoflurane) for murine model in the revised Methods section.

Bibliography.

Comment #13: There are about 9 cited works authored by the corresponding author, Dr. de La Fuente-Nunez, in the Bibliography. We didn't check it to the other authors involved on the group. These citations correspond to 24% of the total references. Although these references are relevant, we do not know if this Journal has any limit for this type of citation. Please check it.

We appreciate the reviewer's observation regarding the proportion of citations. All the references included are directly pertinent to this study and were added to provide comprehensive context and support for our findings, as previously highlighted by the reviewers. Following the revisions, we have expanded and diversified the reference list to include additional relevant studies from a broad range of sources.

Reviewer #4 (Remarks on code availability):

The correspondent author is registered in this site as a member. His last activity was Jan, 30, 2024. Another co-author is also registered. The site provides several options and explanations. It seems accessible, but I didn't explore it.

We appreciate the reviewer's feedback regarding the accessibility of our APEX model repository. To further improve usability, we have updated APEX by adding detail tutorial for installation and running (<https://gitlab.com/machine-biology-group-public/apex/-/blob/main/README.md>). These updates ensure that researchers can more easily reproduce our results and we are committed to the ongoing maintenance and support of these software.

Reviewer #5 (Remarks to the Author):

We are grateful to Reviewer #5 for their time and effort in co-reviewing our manuscript as part of the *Nature Communications* initiative. We greatly appreciate the thoughtful feedback provided and acknowledge the value of initiatives that promote the training and recognition of Early Career Researchers in the peer review process. Thank you for your valuable contribution to the evaluation of our work.

RESPONSES TO REVIEWERS' COMMENTS:

NB. Original comments are in italics and our answers in normal typeface. All additions to the text are colored in red in the modified version of the manuscript.

Reviewers' Comments:

Reviewer #2:

Reviewer #2 (Remarks to the Author)

The authors addressed the raised concerns in a detailed manner and improved the quality of the manuscript by adding the hemolytic activity assays and the additional clarifications on methodological choices.

We are grateful for the reviewer's positive assessment and for the constructive feedback that strengthened our manuscript.

A few minor remarks should be taken into account:

1. With regards to the added explanation about why peptides with length ≤ 7 were excluded from consideration (Page 24, lines 729-731). It would be beneficial if the authors added a reference to support this claim, just as they did in their response to reviewers.

We thank the reviewer for pointing this out. We have now added references 48 and 49 to substantiate the exclusion criterion.

2. I believe the authors inadvertently left a sentence partially unfinished in lines 725-726 ("You also").

We thank the reviewer for noticing this. We have now completed the sentence (Page 23, lines 660-661 in the revised manuscript).

(Remarks on code availability)

With regards to code availability and the ease of setting up the conda environment. The amendments the authors undertook significantly improved the process of setting up a computational environment. However, I believe the equality signs necessary for specifying version numbers are missing from the pip command. More specifically, this command:

"`pip install torch1.11.0+cu113 torchvision0.12.0+cu113 torchaudio==0.11.0 --extra-index-url https://download.pytorch.org/whl/cu113”`

will give an error and it should be rewritten as:

“pip install torch==1.11.0+cu113 torchvision==0.12.0+cu113 torchaudio==0.11.0 --extra-index-url <https://download.pytorch.org/whl/cu113>”;

We thank the reviewer for pointing this out, we have updated both the manuscript and the repository README file with the appropriate installation command:

```
pip install torch==1.11.0+cu113 torchvision==0.12.0+cu113 torchaudio==0.11.0 --extra-index-url https://download.pytorch.org/whl/cu113
```

Moreover, given that the code runs only on a CUDA-capable device, I would encourage the authors to include said information in the repository’s README file as well. Also, it is not necessary to put the commands for setting up a conda environment in the manuscript (Page 24); however, I don’t mind them being there, so the authors can decide what they prefer.

We thank the reviewer for the suggestion. We have added a note in the README that the code requires a CUDA-capable GPU. The conda environment instructions remain in the **Methods** section to facilitate reproducibility.

Reviewer #3:

We thank Reviewer #3 for their time and effort invested in co reviewing our work as part of the *Nature Communications* peer review training initiative.

Reviewer #4:

All suggestions were addressed by the authors, and these modifications significantly enhance the technical and scientific quality of the study.

However, we would like to highlight two points:

1. We emphasize the importance of including compounds that reduce the adhesion of antimicrobial peptides to plastic surfaces (such as Tween-80, for example) in the culture medium in future studies. Additionally, the use of culture media other than Mueller-Hinton should be discouraged. The authors justify the use of LB broth due to its complex biochemical composition. While the described characteristics are indeed valid, Mueller-Hinton broth remains the most widely recommended medium worldwide. The use of rich media such as LB, BHI, and Nutrient broth may lead to false-negative results due to interactions between antimicrobial compounds and components of these media, which can reduce the bioavailability of the antimicrobial agents.

Moreover, the justification regarding the interaction with starch in Mueller-Hinton broth is not convincing, since this medium is specifically recommended for testing cationic antimicrobial peptides and glycopeptides that are already in clinical use, such as polymyxins and daptomycin. Despite the interaction with starch, this effect is still less significant than the interaction with lipid residues present in rich media like LB, Nutrient, and BHI. Furthermore, Mueller-Hinton is a standardized medium in terms of lipid, protein, and carbohydrate content, unlike rich media that contain non-standardized extracts. The lack of standardization in rich media leads to significant batch-to-batch variability, which is not the case for Mueller-Hinton, known for its consistency—another reason why it was standardized and remains the preferred medium. Despite its limitations, Mueller-Hinton broth is still the most appropriate and reliable medium. Therefore, we strongly recommend that future studies align with international guidelines for antimicrobial susceptibility testing.

We appreciate this detailed guidance. While LB broth enabled us to probe a broad biochemical context in the present study, we recognize the standardization and lower adsorption profile of Mueller Hinton broth. In future work we will (i) prioritise MH broth in accordance with CLSI/EUCAST guidelines, and (ii) incorporate non-ionic surfactants such as Tween-80 to minimize peptide adsorption to plastic surfaces.

2. Considering the potential activity of the peptides on ion channels, which was thoroughly emphasized by a reviewer, the authors utilized a predictive program to assess their activity on these channels, as suggested. They stated in the manuscript:

“Moreover, venom-derived compounds may possess toxicity via modulation of ion channels. Our ion channel modulation predictions indicate that nearly 40% of the VEPs identified here do not affect the potassium channel (Supplementary Table 4). Further experiments are warranted to confirm these findings, and additional optimization steps may be required to assess toxicity.”

This simulation included Ach channels, sodium channels, calcium channels, and potassium channels. It was shown that 40% of the selected peptides were not predicted to affect these channels. However, the authors highlighted that the peptide UniProtKB-7 did not modulate potassium channels. It is important to note that 60% of the peptides were predicted to modulate potassium channels, and among the three peptides tested in vivo, UniProtKB-7 was the least active, while the other two, Cono-Server14 and Arachnoserver-5, did show predicted modulation of potassium channels. Although the topical application of these peptides is an attenuating factor in terms of potential systemic effects, the authors should underscore the necessity of conducting electrophysiological assays on specific ion channels to rule out potential toxicity, especially if systemic administration is considered in future applications. Given that potassium channels represent a large family with crucial roles in numerous biological processes, the possibility of absorption—even with topical use over extended periods—could result in side effects. This

represents a critical issue that should be addressed, potentially in a future study focusing on the pharmacological activity of the peptides.

We agree. We now explicitly state in the **Limitations of the study** (Pages 11–12, lines 342–352) that comprehensive electrophysiological profiling, particularly against diverse K⁺ channel subfamilies, will be essential before systemic administration is contemplated. Although beyond the scope of this paper, this work is planned for the next phase of the project.

Minor: page 24, lines 725-726. Please check the phrase "you also...." it seems incomplete.

We thank the reviewer for noticing this. We have now completed the sentence (Page 23, lines 660-661 of the revised manuscript).

Reviewer #5 (Remarks to the Author):

We are grateful to Reviewer #5 for co-reviewing this manuscript and for contributing to the Nature Communications Early Career Researchers programme.